behaviour/psychology/ecology

personality, plasticity, flexibility, sociality, primate

**Author for correspondence:**
Patrick J. Tkaczynski
e-mail: patrick_tkaczynski@eva.mpg.de;
pjtresearchltd@gmail.com

# Long-term repeatability in social behaviour suggests stable social phenotypes in wild chimpanzees

Patrick J. Tkaczynski[1,†], Alexander Mielke[2,†], Liran Samuni[1,3,4], Anna Preis[5], Roman M. Wittig[1,4,†] and Catherine Crockford[1,4,†]

[1]Department of Human Behaviour, Ecology and Culture, Max Planck Institute for Evolutionary Anthropology, Leipzig, Germany
[2]School of Anthropology and Museum Ethnography, University of Oxford, Oxford, UK
[3]Department of Human Evolutionary Biology, Harvard University, Cambridge, MA, USA
[4]Taï Chimpanzee Project, Centre Suisse de Recherches Scientifiques, Abidjan, Côte d'Ivoire
[5]Wild Chimpanzee Foundation, Conakry, Guinea

PJT, 0000-0003-3207-2132; AM, 0000-0002-8847-6665;
RMW, 0000-0001-6490-4031; CC, 0000-0001-6597-5106

Consistent individual differences in social phenotypes have been observed in many animal species. Changes in demographics, dominance hierarchies or ecological factors, such as food availability or disease prevalence, are expected to influence decision-making processes regarding social interactions. Therefore, it should be expected that individuals show flexibility rather than stability in social behaviour over time to maximize the fitness benefits of social living. Understanding the processes that create and maintain social phenotypes requires data encompassing a range of socioecological settings and variation in intrinsic state or life-history stage or strategy. Using observational data spanning up to 19 years for some individuals, we demonstrate that multiple types of social behaviour are repeatable over the long term in wild chimpanzees, a long-lived species with complex fission–fusion societies. We controlled for temporal, ecological and demographic changes, limiting pseudo-repeatability. We conclude that chimpanzees living in natural ecological settings have relatively stable long-term social phenotypes over years that may be independent of life-history or reproductive strategies. Our results add to the growing body of the literature suggesting consistent individual differences in social tendencies are more likely the rule rather than the exception in group-living animals.

†These authors contributed equally to the manuscript.

# 1. Introduction

Consistent individual differences in social behaviour, whether measured as spatial association patterns or rates of direct physical interactions such as grooming and aggression, have been identified across various animal taxa [1–8]. Repeatability, the statistical metric by which such stable differences are quantified, is defined as the proportion of variation in trait expression attributable to between-individual differences relative to the overall variation in trait expression [9]. Therefore, high repeatability in a trait, such as social behaviour, suggests low within-individual variance and high between-individual variance in the trait's expression [9,10]. Repeatability in social behaviour thus implies that group-living individuals show individual and stable tendencies in solving social problems and interacting with other group members.

Social integration and bonds are often associated with positive fitness outcomes [11–21]; however, maintaining these relationships incurs certain costs, such as the energy and time dedicated to socializing [22], or increased exposure to pathogens from social partners [23,24]. Where observed, repeatability in social phenotypes suggests most individuals have consistent social strategies despite the apparent advantages of changing those strategies when socioecological settings and individual internal state vary [25]. For example, short-term strategies of social interaction avoidance would be adaptive during periods of disease outbreak [26]. Similarly, during periods of social upheaval, such as instability in dominance hierarchies, individuals may change how they distribute their affiliative or aggressive social investment to reinforce key relationships or dominance position, respectively [27,28]. Variation in resource availability (e.g. number of available mating partners or food availability) may also influence time allocation to affiliative social interactions, rates of aggression, social partner choice or general gregariousness [22,29]. Lastly, an individual's physiological state may vary over time, e.g. during pregnancy and/or the rearing of offspring, in turn influencing motivation for social behaviour [30]. This raises the question of why highly stable social phenotypes are so regularly observed across animal taxa.

Repeatability in a trait can manifest through multiple processes. Individual patterns of social behaviour, and thus social phenotypes, may emerge as a consequence of the interaction between genetics and exposure to the physical and social environment during development [31–35]. The 'social niche hypothesis' specifically proposes that consistent individual differences in behaviour arise due to niche specialization to ameliorate within-species and/or within-group competition for resources [36]. Here, an individual adopts a behavioural or social strategy to acquire resources dependent upon its existing competition-related characteristics, such as body size, health and dominance. Consistent between-individual differences in behaviour and sociality may also arise due to behavioural tendencies associated with specific life-history stages [37–39]. Indeed, longitudinal studies suggest human behavioural tendencies and personality may be more labile than previously thought, with shifts in these tendencies predicted by a combination of age-related change or adjustment to particular life events, e.g. marriage [40–42]. Therefore, confirming the durability of stable social phenotypes requires monitoring individuals as their environment, intrinsic state and/or life-history stage changes over time.

The repeatability of social behaviour has been well explored in relatively short-lived species, both in the wild [1,4,5,7] and in more controlled captive settings [6,8]. In such studies, the comparatively short lifespans of the study species allow for monitoring the stability of social tendencies across different socioecological settings, such as across breeding seasons [1], different life-history stages [8], or even across generations [5]. However, quantifying social tendencies in these animals is often limited to quantifying patterns of spatial association [1,4,6,7], either due to the practical challenges of monitoring directed social interactions, or due to such social behaviour being comparatively rare in a species. While studies of patterns of association are clearly important for understanding the evolution of social phenotypes, for many species, navigating the social environment requires a diverse range of affiliative and agonistic behaviours.

Non-human primates (hereafter primates) are interesting species for studies examining social strategies and related behaviour as they are typically highly social, use a range of behaviours to maintain relationships, and are often conspicuous enough to collect detailed behavioural data [43,44]. However, they are also typically long-lived, presenting a challenge to accurately assess repeatability of social behaviour that is independent of socioecological setting, intrinsic state or life-history stage. Recent studies have demonstrated that social network positions in seasonally breeding primates can be stable across breeding seasons [2,45], while long-running field sites are beginning to reveal that social phenotypes are stable over much of the adult lifespan in monkey species [37,46]. Similar long-term field studies of social phenotypes in our closest living relatives, the great apes, are currently lacking. Given that humans are one of the few species to demonstrate substantial changes to social phenotypes [40–42], determining the stability of individual differences in social behaviour in wild apes offers valuable insights into the evolutionary origins of the social flexibility in humans.

In our study, we examined how social behaviour varies between individuals in wild, adult chimpanzees from Taï National Park, using a behavioural dataset spanning an average of 5 years per subject, and up to 19 years for certain individuals, incorporating more than 7500 full-day focal follows. Specifically, we examined whether there were consistent individual differences in rates of affiliation (grooming) and aggression directed to conspecifics, as well as consistent individual differences in association patterns (how many conspecifics with which individuals associated).

Chimpanzees live in multi-female, multi-male groups with a high degree of fission–fusion dynamics [29,47], allowing considerable variation in the availability of social partners, both day-to-day and more broadly over the course of their long lifespans [48]. Chimpanzee societies also feature various types of cooperative behaviour, such as alliance formation, food sharing, group hunting and territorial patrols, which probably affect fitness and for which strong social relationships are required [49–52]. Therefore, these individuals face diverse social environments and important choices regarding their social behaviour, and indeed have been shown to be flexible in individual social decisions [53]. Here, we directly test whether patterns of stable differences in social behaviour are evident over the longer term, i.e. from day to day or year to year.

Chimpanzees form stable social bonds, typically quantified by rates of grooming exchange [49,51,54]. However, individual chimpanzees are also highly strategic and flexible in grooming partner choice, with decisions to groom dependent on factors such as which bystanders are nearby or the rank of available partners [53]. Furthermore, time devoted to grooming in this species can vary depending on demographic factors such as group size [55]. In our study, we specifically tested for consistent individual differences in rates of grooming conspecifics. Therefore, we focused on stability in time allocation to grooming others rather than stability in partner choice or number of grooming partners. Given the high degree of fission–fusion in chimpanzees, and the fact that decisions of when to groom others are highly contingent on party size and composition, we anticipated rates of grooming to have relatively low repeatability compared with either aggression or rates of association (see below).

Both male and female chimpanzees use aggression to solve conflicts of interest with other individuals [28,56]. In terms of dominance structure, when compared with female hierarchies, male hierarchies are dynamic and defined by high male–male competition, and there tends to be considerable reproductive skew towards high-ranking males [57–59]. Although females do change rank, female hierarchies are comparatively stable [56,60–62]. Among Taï chimpanzees, rates of aggression vary according to dominance rank and hierarchical stability, reproductive contexts, and group-level social acts such as hunting and patrols [63,64]. Therefore, as with grooming, we expected a high degree of within-individual variation in rates of aggression towards other individuals. Importantly, unlike our predictions for grooming, we anticipated sex differences in repeatability of aggression rates: as male hierarchies are less stable than those of females, within the time frame of our dataset, we anticipated more males to undergo rank changes compared with females, and therefore, vary more than females in their rates of aggression directed towards other individuals. Furthermore, male, but not female, aggression rates vary with mating competition [28,63], leading to lower repeatability of aggression in males compared with females.

Lastly, we tested for consistent individual differences in patterns of association, i.e. the size of the party within which an individual is typically observed. As has been highlighted, chimpanzee sociality is characterized by a high degree of fission–fusion; this allows individuals to adjust with whom they associate (either specific association partners or specific party sizes) dependent on variation in within-group competition arising from ecological constraints, such as the availability of receptive mating partners or food [47]. Competition will vary both seasonally and in the longer term due changing group sizes or sex ratios, favouring variation in the number of individuals with which to associate. However, social bond partners often consistently associate in the same party as each other [49,54,65]. Furthermore, Taï chimpanzees are considered one of the more gregarious chimpanzee populations (i.e. individuals are likely to associate with all other group members relatively frequently, even within a day), probably as a consequence of low population density, high food resource availability and/or predation pressure [66–69]. Therefore, compared with rates of grooming or aggression, we anticipated higher repeatability for levels of association.

Our study focused solely on adults; there is some evidence that within adulthood, both males and females adjust reproductive, and thus life-history, strategies in relation to age and rank [70,71]. Patterns of cortisol secretion across the chimpanzee lifespan suggest males reduce reproductive effort and competition with increasing age and associated decreases in rank [71]; while for females, age tends to be associated with increases in rank and greater time investment (longer interbirth intervals) in offspring [56,62,70]. By controlling for demographic and intrinsic factors, such as group size, reproductive state, dominance rank and age, we aimed to determine the amount of variation in these types of social behaviour that is most likely attributable to individually stable phenotypes [32]. We were also able to collate our data to identify repeatability in social behaviour on two levels, daily

and yearly, allowing us to control for socioecological variables at different temporal scales. Therefore, identifying social phenotypes that are highly stable, contrary to our predictions, would lend support to theories that these phenotypes either become canalized during ontogeny via processes such as social niche specialization [36], with limited flexibility in adulthood, or are heritable traits.

## 2. Methods

### 2.1. Study groups and data collection

Daily, nest-to-nest, focal follow [44] data have been systematically collected by the Taï Chimpanzee Project, Côte d'Ivoire, since 1992 by a combination of local assistants and researchers [72,73]. We focused on data collected since 1996, when data collection was consistent for behaviours relevant to this study and the control factors included in the models (see Statistical analyses). These data include observations of adult (older than 12 years) males and females from three fully habituated communities of chimpanzees: North (1996–2016), South (2002–2016) and East (2012–2016). The order in which subjects were sampled was pseudorandomized, with efforts made to ensure all adults had at least one daily focal sample each month.

We collated these data to identify repeatability in interaction rates on two levels: daily and yearly (with year ending on 31 August), allowing us to control for socioecological variables at different temporal scales. For each of these levels, we restricted the dataset to individuals with regular focal follows to ensure that the data were sufficient to capture their typical social behaviour. The criteria for the inclusion of individuals were as follows:

— For the analysis on the **daily level**, we included focal follow days of adult individuals that lasted at least 3 h, and included individuals for whom at least 10 focal follow days were available, resulting in a dataset of 70 individuals (45 females, 25 males) and 7615 individual focal follow days (individual mean = 109 days, max = 413 days). The 3 h cut-off value was chosen as observers would sometimes lose sight of focal chimpanzees during follows and change the focal individual. Changes of subjects did occur at different times of the day, so shorter observations would not be biased towards specific behaviour. The majority of observations (81%) were in excess of 8 h; the mean focal sample length was 9.4 h (±1.9 h s.d.).
— For the analysis on the **yearly level**, we included adult individuals who were followed at least nine times in the time period from 1 October to 30 September and were present in their group for all this period, and who fulfilled these criteria at least in 3 years. This resulted in a dataset of 45 individuals (24 females, 21 males) and 272 years (individual mean = 6 years, max = 15 years).

The number of years or days per subject of the study are detailed in table 1.

### 2.2. Social phenotype

On both the daily and the yearly level, we focused on three social variables: grooming, aggression and association. We calculated variables from the perspective of the focal individual and with the focal as the actor (in case of the interactions) to ensure that they represent the individual's social propensity as much as possible.

For **grooming**, we extracted the time (in minutes) focal individuals spent grooming adult partners. We focused on grooming given to others rather than overall time grooming, i.e. including grooming received, as this would reflect a tendency to attract grooming partners rather than an individual tendency to groom. This variable included all grooming given to other adults by the focal, including unidirectional, mutual (both giving and receiving grooming at once) or polyadic (grooming one other individual but potentially being grooming by multiple other partners) grooming.

For **aggression**, we extracted the number of directed aggressive interactions in which the focal individual was the aggressor (aggression included non-contact aggression such as chases, charges and threats; and contact aggression). Both these variables were calculated the same for the daily and yearly level.

We calculated **association** differently for the two levels:

— On the *daily* level, we calculated the total unique adult individuals with whom the focal associated on a given day, with association defined as being observed within the same 'party' (defined as all individuals within visual contact of the observer—usually less than 35 m) during that day.

**Table 1.** Total numbers of observations for focal chimpanzees at the daily and yearly level. We show the observation length per individual for each level of data aggregation. We also show the time range (i.e. the first and last years) in which individuals appeared in the dataset. Individuals may not be observed consistently within this range and, therefore, not every day/year is included in the analysis (see Methods for inclusion criteria).

| individual | sex | days | years | range | individual | sex | days | years | range |
|---|---|---|---|---|---|---|---|---|---|
| 1 | M | 146 | 4 | 2011–2015 | 36 | F | 76 | 3 | 2011–2015 |
| 2 | F | 130 | 7 | 2001–2009 | 37 | F | 413 | 11 | 1996–2015 |
| 3 | F | 33 | — | 2011–2014 | 38 | F | 12 | — | 2015–2015 |
| 4 | F | 387 | 10 | 1996–2015 | 39 | F | 377 | 14 | 1996–2015 |
| 5 | M | 95 | 3 | 2001–2004 | 40 | M | 181 | 10 | 1999–2008 |
| 6 | F | 18 | — | 2008–2012 | 41 | M | 152 | 5 | 2008–2015 |
| 7 | M | 33 | — | 1996–1997 | 42 | F | 98 | 6 | 2001–2009 |
| 8 | F | 47 | 3 | 1996–1999 | 43 | F | 59 | 3 | 2011–2015 |
| 9 | F | 16 | — | 2012–2015 | 44 | F | 49 | 3 | 2013–2015 |
| 10 | F | 56 | 3 | 1996–1999 | 45 | F | 327 | 15 | 1996–2015 |
| 11 | F | 25 | — | 2012–2013 | 46 | F | 70 | 3 | 2011–2015 |
| 12 | F | 64 | 3 | 2011–2015 | 47 | M | 104 | 4 | 2011–2015 |
| 13 | F | 19 | — | 2001–2002 | 48 | F | 11 | — | 2014–2015 |
| 14 | M | 68 | 4 | 2012–2015 | 49 | M | 88 | 3 | 2013–2015 |
| 15 | F | 202 | 11 | 1996–2006 | 50 | M | 23 | — | 2010–2013 |
| 16 | M | 149 | 4 | 2011–2015 | 51 | F | 28 | — | 2001–2007 |
| 17 | F | 13 | — | 2014–2015 | 52 | F | 34 | — | 2014–2015 |
| 18 | F | 11 | — | 2014–2015 | 53 | M | 266 | 8 | 2001–2009 |
| 19 | M | 131 | 6 | 2002–2008 | 54 | M | 101 | 3 | 2013–2015 |
| 20 | F | 106 | 6 | 1996–2001 | 55 | F | 298 | 13 | 2001–2015 |
| 21 | F | 18 | — | 2014–2015 | 56 | F | 130 | 5 | 2007–2015 |
| 22 | M | 118 | 3 | 2011–2015 | 57 | M | 27 | — | 2008–2009 |
| 23 | F | 76 | 4 | 2011–2015 | 58 | F | 14 | — | 2013–2015 |
| 24 | F | 251 | 11 | 2001–2015 | 59 | M | 83 | 4 | 2008–2011 |
| 25 | M | 126 | 5 | 2010–2015 | 60 | F | 188 | 8 | 1996–2005 |
| 26 | F | 138 | 3 | 2001–2015 | 61 | F | 59 | — | 2011–2015 |
| 27 | F | 20 | — | 2001–2002 | 62 | F | 40 | — | 2001–2009 |
| 28 | M | 193 | 8 | 2001–2009 | 63 | M | 92 | 3 | 2013–2015 |
| 29 | F | 166 | 6 | 2003–2015 | 64 | M | 245 | 10 | 2006–2015 |
| 30 | M | 190 | 7 | 2009–2015 | 65 | F | 33 | — | 2013–2015 |
| 31 | F | 124 | 5 | 2001–2009 | 66 | F | 15 | — | 2011–2015 |
| 32 | F | 51 | 3 | 1996–1999 | 67 | M | 21 | — | 2001–2002 |
| 33 | M | 84 | 3 | 1996–1999 | 68 | F | 40 | — | 2001–2005 |
| 34 | F | 13 | — | 2001–2002 | 69 | F | 79 | — | 2001–2009 |
| 35 | M | 235 | 8 | 1996–2003 | 70 | M | 230 | 8 | 2001–2009 |

— On the *yearly* level, all individuals will probably associate with all available social partners within this timeframe, making it impossible to differentiate between individuals in rates of association. Therefore, we instead calculated the total *strength* of connections of individuals within social networks of association [74]. Here, we created matrices of association, i.e. counts of how frequently dyads of individuals were observed in the same party in one year; then weighted these counts by

the observation effort of each individual within the dyad using the simple ratio index [74]. An individual's strength within such a network is the sum of its weighted ties to all other adult individuals, i.e. how often an individual tends to be observed with other individuals given overall observation effort for all individuals. We standardized association strength by dividing it by group size.

Our yearly association variable is a social network metric; such metrics are inherently non-independent as they are derived from the emergent properties of the whole network and association indices are shared between a dyad within a matrix [75]. Therefore, association indices have to be considered in the context of an appropriate null model [75]. We conducted permutation analyses to confirm that associations were different than would be expected by chance. For this analysis, we generated 1000 permutations of observed parties in which the number of individuals per party was kept constant; subsequently generated parties that had the same party membership as the observed party were included in the analysis to account for autocorrelation [76]. The overall number of times individuals appeared was not held constant, as we were interested in whether they would be more central (higher association) than expected. We subtracted the mean strength of individual's connections with all other group members arising from the permutations from the observed strength, effectively creating an index of association (above 0 = more connected than would be expected in that group in that year; below 0 = less connected than expected).

## 2.3. Statistical analyses

All data were prepared and analyses were conducted using R 3.6.1 statistical software [77], with generalized linear mixed models (GLMMs) fitted using the 'lme4' package [78]. Models were fitted separately for males and females, for the dependent variables of aggression, grooming and association, and for the daily and yearly levels of data aggregation, resulting in 12 models. Within each model, we controlled for several socioecological factors or variation in intrinsic state and life-history stage or strategy.

   Socioecological variables:

— As the **sex ratio** of adult individuals in a group can influence social behaviour, we calculated the adult sex ratio (number of males/number of females) as the mean of daily sex ratio of each group for the yearly period (for the yearly level) or of those individuals present in the focal individual's party during the day (for the daily level).
— As interaction rates could be determined by the **number of available partners** (on the daily level) or mean daily **group size** (on the yearly level), we included these variables in the models. The yearly group size variable was standardized within groups (z-score transformation based on the mean and standard deviation of group's size across the study period); otherwise, there would be complete separation between the group identity and group size (as North group was considerably smaller than the others).
— We accounted for **seasonality** effects in the daily level analyses by including the radians (sine and cosine) of Julian date as a control variable [79]. This proxy seasonality measure controls for the departure from a uniform distribution of a variable (here social behaviour rates) over time [79,80].

State/life-history variables:

— The **age** of all individuals was either known (for individuals born during or post habituation) or was estimated in the beginning of the habituation period by experienced observers using established indicators of ageing in wild chimpanzees [81]. Of the 70 subjects, 39 had estimated ages. Age ranged from 12 years (cut-off value) to 52 years, and the value assigned was either for the day (daily level) or the end of the yearly period (yearly level). We included age as a squared term in all models to account for potentially nonlinear developmental patterns, e.g. individuals in their prime being more aggressive than younger and older individuals.
— **Dominance rank** was calculated using a modification of the Elo rating method [82,83] based on unidirectional pant grunt vocalizations within each sex and standardized between 0 and 1 within each group. Individual Elo ratings reflect the likelihood to 'win' a dominance interaction over an opponent based on previous dominance interactions; traditional Elo approaches use a fixed starting value for individuals and a fixed value $k$ for the initial changes in ratings based on outcomes of dominance interactions. The Elo modification used here uses a maximum-likelihood estimation to optimize $k$ and allow individuals to enter the hierarchy with different starting values [82,83].

— For the daily level, rank was extracted for each day of data an individual was observed, providing a rank value for each day of observation for each individual. For the yearly value, we extracted the mean daily rank for each year, providing a single rank value for each year for each individual.

— For females, as **reproductive state** and **infant age** influences association patterns and sociality [53,67,68,84,85], on the daily level, we included a factor with five levels: maximum tumescence of sexual swellings, mother with a newborn infant (below three months of age), mother with an unweaned offspring (below 4 years of age), mother with weaned but immature offspring, or none of the above. In instances where females had weaned or unweaned offspring but were also maximally tumescent or had a newborn infant, they were classified as maximally tumescent or with a newborn infant, as these reproductive states were anticipated to have a greater effect on behaviour than the presence of older offspring [79–83]. On the yearly level, we included a factor with three levels: whether they had no offspring throughout the year, had a newborn at any point, or unweaned offspring. As with the daily measure, the presence of a newborn was prioritized over the presence of unweaned offspring in this categorization. On the daily level for the analyses concerning males, we included the *number of females with maximally tumescent sexual swellings* on that day as a variable.

Other control variables:

— For count models at both daily and yearly levels of analysis, we included an offset term for focal **observation time** (in hours; log-transformed) to account for differences in base rates of how often individuals could be observed interacting.
— All models included a random effect of '**individual identity**' to account for repeated measures of individuals and to calculate the contribution of individual identity to variation in social behaviours.
— All models included a random effect of '**group-year**': Taï chimpanzees show group-specific interaction rates [86] that may change over time. This term accounts for overall differences that could be influencing how social individuals are.

## 2.4. Individual model structures

For **aggression** (both on the daily and yearly level, and males and females), we fitted GLMMs with Poisson error structure and log link [87], using the number of bouts per day/year as outcome variable, and having a log-transformed offset term for observation time to control for differences in observation effort [88]. Because of large numbers of zeros for female daily aggression, we fitted a negative binomial model with the same specifications. For the daily level, we included age as a quadratic term, dominance rank, available partners (daily level) or group size (yearly level), sex ratio, and group identity as fixed-effect predictors. For the daily level, we also included the number of females with maximum swellings and the cosine and sine functions to capture seasonal effects. We included the random intercept of group-year, to account for the fact that data collected in the same time period in the same group are not independent, and the random intercept of individual identity. We also included the random slopes for rank, available partners and sex ratio in individual identity, and rank and available partners in group-year [89].

For **grooming**, on the yearly level, we fitted LMMs with Gaussian error structure for the minutes per hour grooming rate. As the rate did not include many zeros, but was strongly left-skewed, we used the log-transformed hourly grooming rate as outcome variable. On the daily level, due to zero inflation for the grooming rate, we fitted negative binomial GLMMs with the grooming duration (rounded to full minutes) as outcome variable and observation time as an offset term. The fixed effects and random effect structure were the same as described for the aggression models.

For **association** on the daily level, we fitted GLMMs with Poisson error structure, using the number of unique adult individuals that the focal associated with on the day as outcome variable, and the number of available partners in the group as offset term. The fixed effects were the same as described for aggression, bar the removal of the available partners from the fixed effects. For association on the yearly level, we used the difference between observed and expected association strength as outcome variable, fitting LMMs with Gaussian error structure, and the same fixed effects as described above.

For all models, we z-standardized continuous predictor variables to facilitate interpretation [90]. We compared the full models including all predictors and random intercepts and slopes with a null model including all fixed and random effects except the random intercept and slope for the individual identity [1,91], to test whether stable individual differences had an impact on interaction distributions. To examine the strength of the observed effect of individual identity only [92], we applied the 'r.squaredGLMM'

function of the 'MuMIn' package [93] on a reduced model containing only the fixed effects and the random intercept and slopes for individual identity. We assumed that the variance in social behaviour that can be attributed to intra-individual stability can be captured by subtracting the variance explained by the fixed effects alone (marginal effect size) from that explained by the fixed effects and the individual random effect structure (conditional effect size) [9,92,94]. Thus, for both the daily and the yearly level, both sexes, and all three types of social behaviour, we report whether the random effect significantly improved predictability, the marginal and conditional effect size, and their difference.

To avoid problems due to multicollinearity, we established the variance inflation factors (VIF) of each model [95] using a simple linear regression of the fixed effects and applying the 'vif' function of the 'car' package [96]. The maximum VIF for any model was 3.13, indicating that collinearity was not an issue. For all Poisson models, we tested for overdispersion; models with high overdispersion were changed to negative binomial models, after which overdispersion was no longer an issue in any model. We checked non-normality of residuals by creating qq-plots and homoskedasticity by plotting squared residuals against fitted values, which did not reveal substantial problems.

# 3. Results

In all models, except for female daily aggression, including the random intercept and slopes of individual identity significantly improved model fit, indicating that for most social behaviours involved in this study, chimpanzees showed inter-individual differences that were consistent over time and could be detected on the daily and yearly level.

We report the effect sizes attributed to the individual random effects and the comparison of the full model containing the random effect and the null model without it for the different models in table 2 and figure 1. The variation explained by the random effect of individual identity varied between 0.05 and 0.61. Inter-individual differences in females tended to be more pronounced on the yearly level (mean = 0.51) than on the daily level (mean = 0.20), while differences were minimal between levels for males (yearly mean = 0.30, daily mean = 0.27).

In general, inter-individual differences between females were more stable than between males on the yearly level (female yearly mean = 0.51 versus male yearly mean = 0.30), while the two sexes did not differ strongly on the daily level (female daily mean = 0.20 versus male daily mean = 0.27). Thus, after accounting for age, dominance rank, reproductive state (for females), group-level sex ratio and group size, stable inter-individual differences in yearly interaction rates were more pronounced in females than in males, except in grooming. Daily aggression rates in female chimpanzees showed little variation between individuals, mainly because aggression was rare, with a maximum daily number of aggressions of six bouts, leaving little room for deviation.

For grooming and aggression, the residuals of individuals that were represented in both the yearly and daily models were highly correlated (table 2), while this was not the case for association levels, where different outcome variables were used. Figures 2 and 3 illustrate inter-individual variation in grooming, aggression and association at the daily and yearly levels in male and female chimpanzees, respectively.

# 4. Discussion

Our study reveals that multiple types of social behaviour are repeatable in wild chimpanzees over several years. Given the results of our study and work conducted in taxa ranging from insects [5] to fish [6,7] to birds [1] to primates [2,45,97], consistent individual differences in social phenotypes seem the norm for group-living animals regardless of the degree of social complexity or environmental heterogeneity. Contrary to our expectations of low behavioural repeatability based on the social complexity of the social system in chimpanzees, the repeatability estimates for social behaviour in our study were comparable to an average repeatability estimate of behavioural traits across numerous animal taxa formerly generated by a meta-analytical study ($R = 0.37$; [10]). Using a comprehensive dataset, we were able to control for temporal, seasonal and demographic changes across the long-term study period that may influence variation in social behaviour, and thus limit pseudo-repeatability [32]. As such, our method suggests that our repeatability estimates reflect stable social behaviour phenotypes that may be somewhat independent of variation in socioecological settings, intrinsic state and life-history or reproductive strategy during adulthood in wild chimpanzees.

**Table 2.** Repeatability coefficients of chimpanzee social behaviours. Effect sizes and results of full versus null model comparisons for the models testing the impact of the individual random intercept and slopes on grooming rates, aggression bouts and association in male and female Taï chimpanzees on a daily and yearly level. The correlation between the two levels was conducted on the random intercept residuals of individuals who were present in both datasets.

| | | | marginal $R^2$ (fixed effects variance) | conditional $R^2$ (fixed + random effects variance) | individual $R^2$ (conditional $R^2$ − marginal $R^2$) | full versus null model comparison | | correlation daily–yearly |
|---|---|---|---|---|---|---|---|---|
| | | | | | | $\chi^2_{\text{d.f.}=5}$ | $p$-value | |
| female | grooming | daily | 0.05 | 0.19 | 0.14 | 51.85 | <0.001 | 0.80 |
| | | yearly | 0.18 | 0.63 | 0.45 | 47.18 | <0.001 | |
| | aggression | daily | 0.06 | 0.11 | 0.05 | 6.27 | 0.281 | 0.69 |
| | | yearly | 0.19 | 0.66 | 0.57 | 13.87 | 0.008 | |
| | association | daily | 0.25 | 0.67 | 0.42 | 450.93 | <0.001 | 0.23 |
| | | yearly | 0.11 | 0.72 | 0.61 | 37.74 | <0.001 | |
| male | grooming | daily | 0.13 | 0.34 | 0.21 | 67.65 | <0.001 | 0.93 |
| | | yearly | 0.33 | 0.74 | 0.41 | 28.72 | <0.001 | |
| | aggression | daily | 0.30 | 0.54 | 0.24 | 150.69 | <0.001 | 0.55 |
| | | yearly | 0.67 | 0.93 | 0.26 | 202.32 | <0.001 | |
| | association | daily | 0.47 | 0.83 | 0.35 | 71.08 | <0.001 | 0.16 |
| | | yearly | 0.25 | 0.46 | 0.21 | 9.62 | 0.022 | |

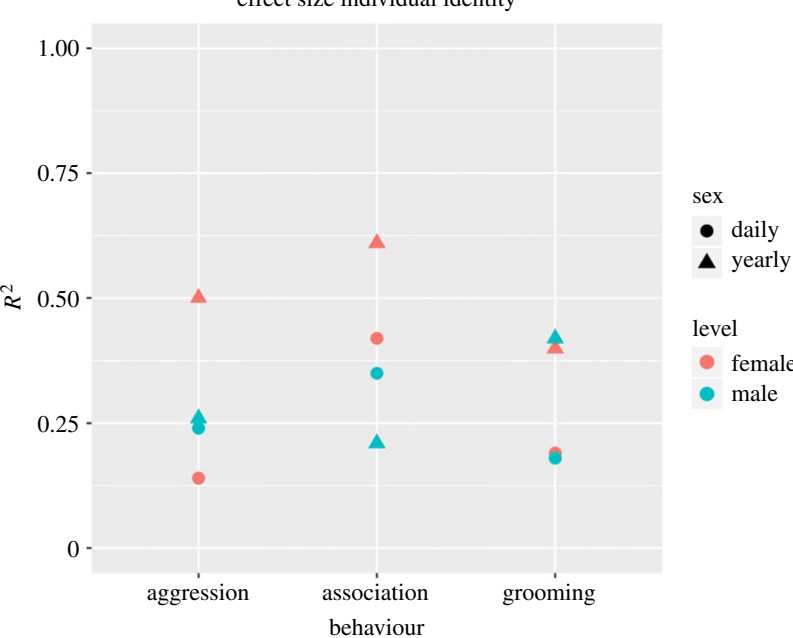

**Figure 1.** Repeatability of chimpanzee social behaviours. Overview of the effect sizes attributed to the individual random effect, depicted by behaviour, sex and timeframe (yearly versus daily) over 20 years of data. $R^2$ refers to individual $R^2$, i.e. the difference between conditional $R^2$ (combined variance of fixed and random effects) and marginal $R^2$ (variance of fixed effect only).

## 4.1. Variation in repeatability between types of social behaviour

We found consistent individual differences in multiple forms of social behaviour. Much research to date on social phenotypes has focused solely on association patterns [1,7], or on single forms of dyadic interaction [20,34], with a minority of studies examining consistency in multiple forms of social behaviour [45,98,99]. Our study shows that consistent individual differences in social behaviour extend to patterns of aggression and grooming. Both aggression and grooming involve direct, typically physical interactions with other group members. In chimpanzees and other species with similar social behaviour and structures, aggression and grooming influence rank acquisition [53,56,83,100,101], disease transmission [23,24,102], social bond formation and maintenance [49,51,54,65], and group cohesion for territorial defence [52,64,103,104]. These factors can have direct and indirect fitness implications in chimpanzees, e.g. frequent incursions from neighbouring groups is associated with reduced infant survival and thus reduced adult reproductive success [105]. Given the potential significance of these social tendencies, understanding how certain individuals come to be more aggressive or affiliative, as well as gregarious, than others, requires further empirical exploration.

## 4.2. Sex differences in repeatability of social behaviour

For all three types of social behaviour, there were interesting differences in repeatability between the sexes. In chimpanzees, there are well-established sex differences in how individuals compete for positions within dominance hierarchies [63,106–108]. Male aggression rates were anticipated to be more variable in relation to more unstable dominance positions compared with females. In line with our predictions, sex differences in the maintenance of dominance hierarchies were reflected in our repeatability measures: males, on average, tended to have lower repeatability for rates of aggression than females (although see below for how the level of data aggregation affected female repeatability estimates).

The results for our more prosocial behavioural measures, grooming and association, highlight how social settings can constrain social choices and repeatability outcomes. As predicted, repeatability in association was higher than for the other two types of social behaviour. Females had higher repeatability estimates than those calculated for males for association. However, for both sexes, within-individual variation appeared much lower for association compared with the other social measures. Taï females are considered more gregarious than females in other chimpanzee populations [67,68], which may be due to predation pressure that is absent in other populations [66]. Taï females do

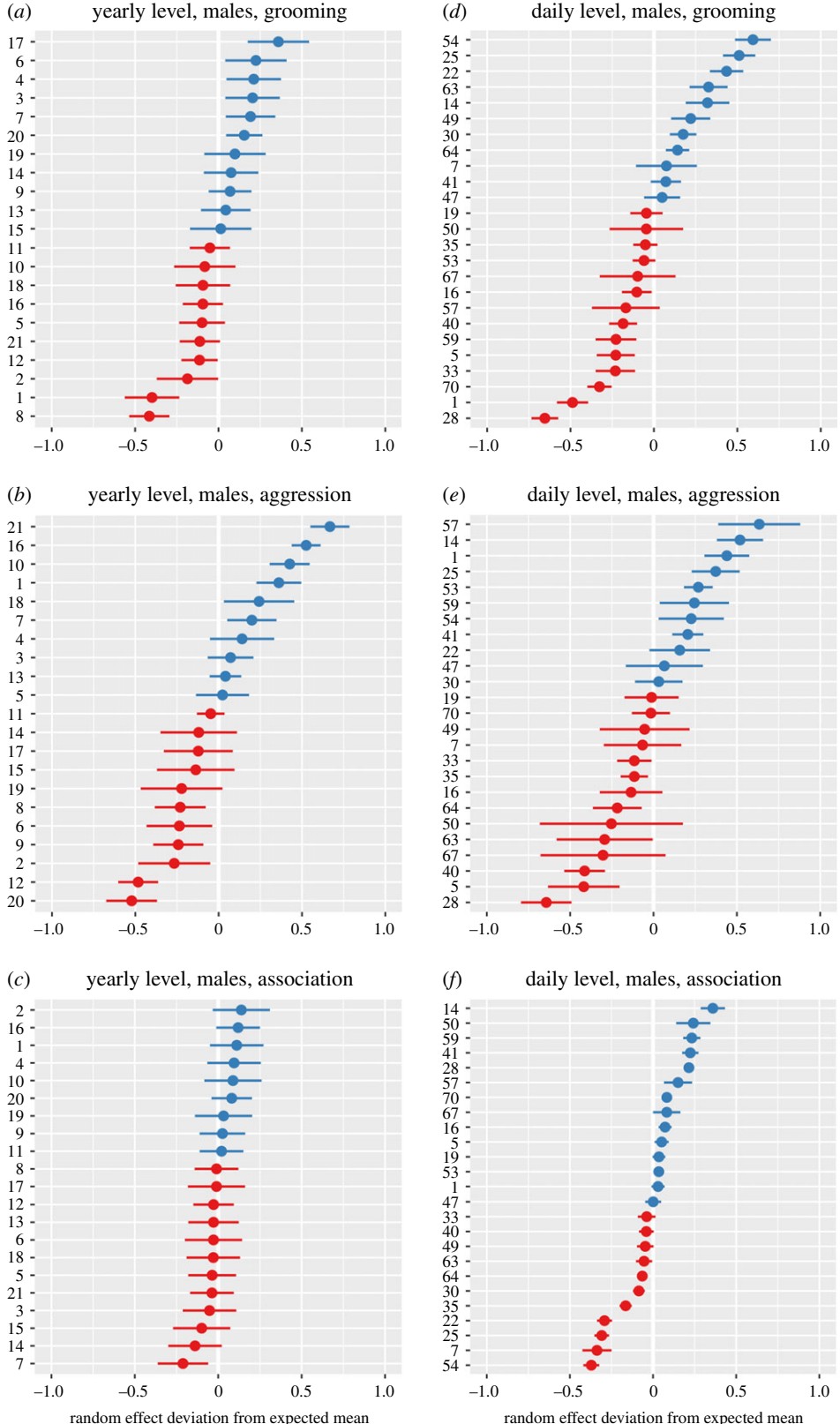

**Figure 2.** Inter-individual variation in grooming, aggression and association in male chimpanzees. Plots include random effect coefficients for individual identity from models examining variation in grooming, aggression and association in males at both the daily (a–c) and yearly (d–f) level of aggregation. The x-axis indicates the mean and variation of individual random effect coefficients; the y-axis indicates individual identities. Blue individuals: higher than average expression of the variable of interest (grooming, aggression and association), accounting for all fixed and random effects within the model, red individuals: less than average expression of the variable of interest.

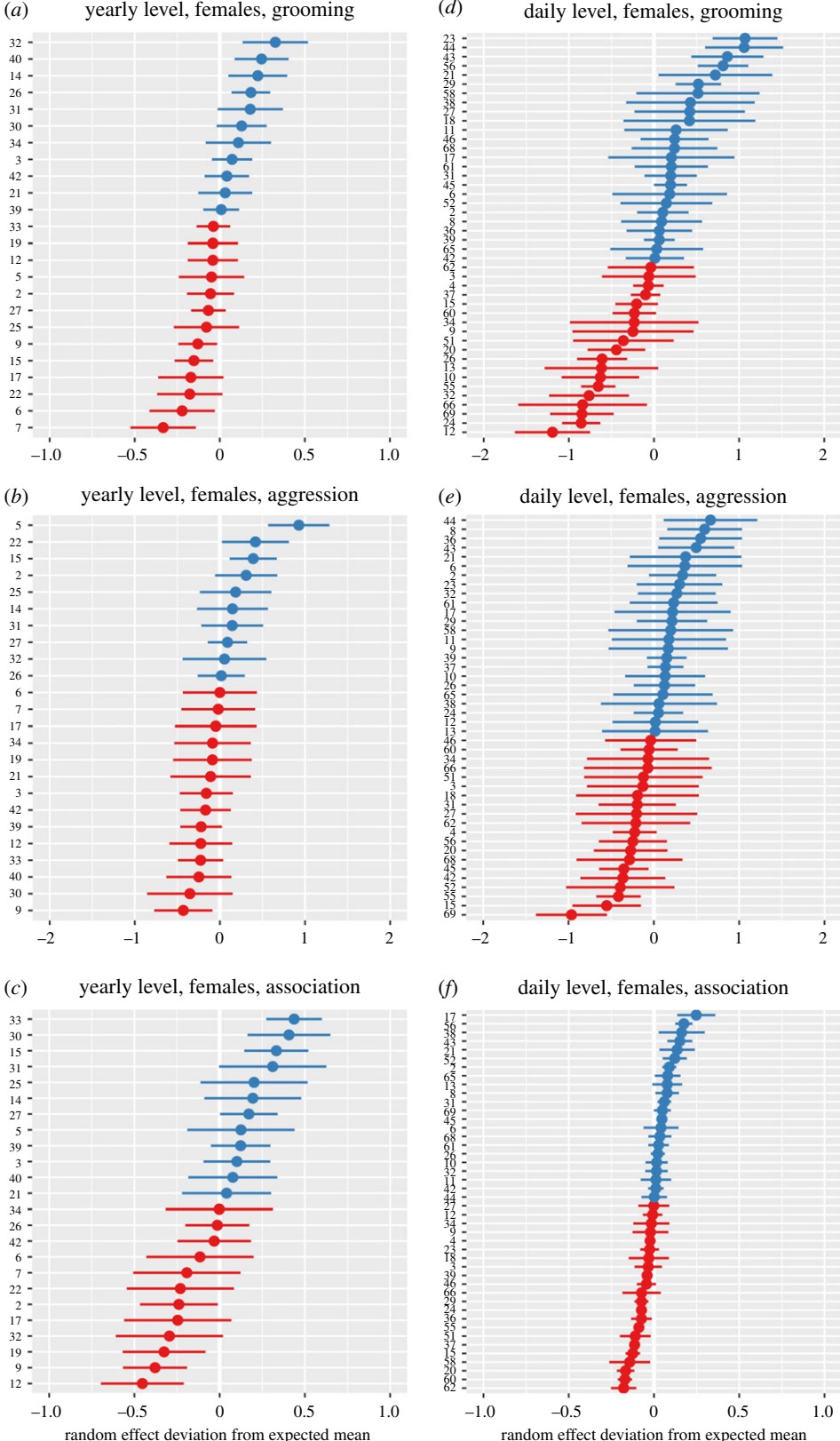

**Figure 3.** Inter-individual variation in grooming, aggression and association in female chimpanzees. Plots include random effect coefficients for individual identity from models examining variation in grooming, aggression and association in females at both the daily (*a–c*) and yearly (*d–f*) level of aggregation. The *x*-axis indicates the mean and variation of individual random effect coefficients the *y*-axis indicates individual identities. Blue individuals: higher than average expression of the variable of interest (grooming, aggression and association), accounting for all fixed and random effects within the model, red individuals: less than average expression of the variable of interest.

adjust their rates of association to resource abundance and avoid parties with sexually receptive females [67,68]. However, in a highly gregarious population with small group sizes and high predation risk, there may be both limited opportunities and advantages to frequently varying rates of association. For males, the risks of predation may be lower (they are both slightly larger and do not travel with dependent offspring), allowing more flexibility in association partners.

We did not observe sex differences in the repeatability estimates for grooming, and within-individual variation in this behaviour again seems comparable for each sex. While grooming forms an important component of social bond formation [109], chimpanzees also make contingent grooming choices based on a range of parameters, such as audience, partner rank or context (e.g. reconciliation after an aggression) [53]. Therefore, when compared with association rates, there may be substantially higher flexibility in social decisions for types of behaviour involving directed interactions such as grooming, and indeed aggression. As such, these types of behaviour are perhaps more reflective of individual social tendencies compared with association rates.

## 4.3. Repeatability and data aggregation

Our results highlight the impact of different temporal levels of data aggregation in repeatability analyses, with implications for future research in this field. Repeatability was generally lower in the daily versus the yearly level across the three types of social behaviour, with implications for data aggregation and interpretation in repeatability analyses. Our motivation to aggregate data at two separate levels was to be able to include control variables at different temporal scales. The daily measures incorporate more data points with potentially fewer social interactions within them compared with the yearly measures. This may result in more random error and thus lower repeatability estimates for the daily level. However, aggregating data at the daily level allows one to accurately control for day-to-day variation in factors such as number of available social partners or the immediate effects of change in dominance rank when they occur.

The contrast between daily and yearly levels of repeatability was strongest in female aggression, with much higher repeatability estimates in the yearly versus daily data aggregation. Taï females do physically compete over nutritional resources using aggression or displacement of others [56]. The abundance of these resources varies seasonally, therefore, environmental contributors to female aggression are more predictable than those of males (e.g. male–male competition over rank or mating opportunities), and thus, the repeatability in female aggression may be more strongly attributable to individual tendencies. Aggression is a costly behaviour in terms of energetics [110,111] and comes with inherent risks of injury, which could even be lethal in certain chimpanzee populations [112,113]. Therefore, although in the long term certain females are more aggressive than others, on a day-to-day basis, individuals are likely to be selective in when to be aggressive, reducing inter-individual differences and repeatability measures.

The effects of different levels of data aggregation on repeatability estimates requires further investigation. Future studies in this field should account for the trade-off between likelihood to observe a behaviour and the ability to effectively control for the socioecological settings when choosing at what level to aggregate data and thus generate accurate repeatability estimates.

## 4.4. Causes and consequences of repeatable social behaviour

A key goal of our study was to generate highly controlled repeatability estimates to make inferences about the mechanisms underlying the emergence and maintenance of social phenotypes. Contrary to our predictions of highly flexible and unstable behavioural patterns, we observed stable social phenotypes independent of variation in environmental and physiological conditions. Our results do not demonstrate that chimpanzees are incapable of making flexible decisions about who to socialize with and when, but they do show consistent individual differences in how adult chimpanzees socialize, which may suggest early-life canalization of these tendencies or the influence of heritable factors.

The social niche hypothesis proposes that individuals adopt particular social strategies to ameliorate the resource competition based on individual idiosyncrasies that might predict competitive ability, such as body size [36]. Chimpanzees have a protracted immature period, lasting around 10–15 years, during which, young chimpanzees consistently associate with their mothers [114]. That some adult females (which will also typically be mothers) are more aggressive, affiliative and gregarious than others, is potentially significant to offspring development. Given repeatability, mothers consistently expose their

offspring to particular social environments and demonstrate repeatable social decisions [115,116], which may inform the emergence of offspring social phenotypes.

Repeatability in a trait, such as social behaviour, means that the trait can be selected for if the between-individual differences arise from genetic differences [94]. While the heritability of non-social types of behaviour is beginning to be explored [117–119], establishing the heritable component of variation in social phenotypes requires further examination. In order to accurately estimate heritability of social phenotypes, it will be important to also consider environmental effects during ontogeny when phenotypic plasticity is anticipated to be at its highest [120].

In summary, our results add to the well-established literature on the repeatability of behaviour and social tendencies in group-living animals. Our study informs specifically on sex differences in social tendencies in chimpanzees. Furthermore, we show repeatable social behaviour in adults that is largely independent of socioecological variables, suggesting either heritable factors or canalization of these phenotypes during development, which should be confirmed in future studies.

Ethics. All applicable international, national and/or institutional guidelines for the care and use of animals were followed. The study was approved by the Ministère de l'Enseignement supérieur et de la Recherche scientifique, the Ministére des Eaux et Forêts of Côte d'Ivoire and the Office Ivoirien des Parcs et Réserves.

Data accessibility. Data used in this study have been made available at: https://doi.org/10.6084/m9.figshare.12001248.v1.

Authors' contributions. P.J.T., A.M., C.C. and R.M.W. conceived the study; A.M., L.S., A.P., R.M.W. and C.C. collected the data; A.M. performed the statistical analyses. P.J.T. wrote the first draft of the manuscript and all authors contributed substantially to subsequent drafts. All authors read and approved the final manuscript.

Competing interests. We declare we have no competing interests.

Acknowledgements. We thank the Ministère de l'Enseignement supérieur et de la Recherche scientifique, the Ministère des Eaux et Forêts de Côte d'Ivoire and the Office Ivoirien des Parcs et Réserves for permission to conduct the study. We are grateful to the staff of the Taï Chimpanzee Project and the Centre Suisse de Recherches Scientifiques for support. We are indebted to the numerous field staff that have collected the data presented. We thank Andrew Dunn and three anonymous reviewers for their comments and feedback, which substantially improved the manuscript.

Funding. Core funding for the Taï Chimpanzee Project is provided by the Max Planck Society since 1997. P.J.T., C.C. and R.M.W. were supported by the European Research Council (ERC; grant agreement no. 679787). L.S. was supported by the Minerva Foundation. A.P. was supported by the Leakey Foundation. A.M. was supported by the Wenner-Gren Foundation and a British Academy Newton International Fellowship.

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
