## [Reviewer comments · Royal Society Open Science]

Review History

RSOS-200454.R0 (Original submission)

Review form: Reviewer 1

Is the manuscript scientifically sound in its present form?

Yes

Are the interpretations and conclusions justified by the results?

Yes

Is the language acceptable?

No

Do you have any ethical concerns with this paper?

No

Have you any concerns about statistical analyses in this paper?

Yes

Recommendation?

Major revision is needed (please make suggestions in comments)

Comments to the Author(s)

See attached file (Appendix A).

Review form: Reviewer 2**Is the manuscript scientifically sound in its present form?**

Yes

Are the interpretations and conclusions justified by the results?

Yes

Is the language acceptable?

Yes

Do you have any ethical concerns with this paper?

No

Have you any concerns about statistical analyses in this paper?

Yes

Recommendation?

Major revision is needed (please make suggestions in comments)

Comments to the Author(s)

The authors examine consistent between-individual differences in agonistic and affiliative social behavior in wild chimpanzees, using a dense longitudinal sample of individuals living in 3 distinct communities in the Tai forest of Cote D'Ivoire. The authors specifically look at repeatability in daily vs annual levels of aggression, party association, and grooming given. The authors find that individual chimps are generally consistent in their preferences to associate in parties of a particular size and the amount of time they spend in association, though on an annual level males were less consistent than females. Males were also less consistent than females in their tendency to give aggression and grooming.

Recent examinations of consistent inter-individual differences, such as this study, provide important insight into behavioral strategies that deviate from population averages, and that were historically considered noise surrounding species-typical behavior. They also lend important insight into the broader evolutionary and ecological significance of what are usually termed "personalities" in humans. What I believe would strengthen this paper is to contextualize the inter-individual differences better in terms of social strategies that are relevant to chimpanzee social life. I flesh out specific areas for improvement in the following line-by-line comments.

Line 36-38: The length of previous collection time periods alone does not necessarily warrant further research in consistent individual differences. There is quite a lot of emphasis on the size of data set in this study as a measure of its significance. While the data set is impressive for a long-lived animal and certainly hard-won, many species examined for repeatable behavior have been observed over similarly long periods relative to their lifespans (many insects, birds, and fish, e.g. 3 consecutive flocking seasons in great tits that live to 13 years at maximum, Aplin et al. 2015 Animal Behavior). Observation years per individual in this study ranged 6 - 12 years in this study, for an animal that lives perhaps up to 65. Observations in chimps are perhaps more

consistent and even over each year – how might this make for particularly reliable and robust estimations of repeatability?

Intro paragraph starting line 47: Again, what is the significance of repeatable behavior? Currently this paragraph highlights various internal and external stimuli that can cause behavior to vary. While this is true, an interesting aim is to determine whether behavior, which has elsewhere been determined to be adaptive (e.g. affiliation, coalitionary aggression), is flexible to the moment or representative of a more constant trait.

Line 91: I suggesting strengthening the justification for this analysis in chimpanzees. The authors seem to present 2 hypotheses in the introduction, either personality is canalized by early environmental density dependent conditions (social niche), or behavior varies by life history stage and/or socio-ecological condition. If these are two hypotheses the authors wish to test, how are fission fusion dynamics relevant?

Line 145: Clarify, does association mean spatial association/party membership excluding time grooming, or do these 2 measures overlap?

Line 153: I am confused as to the kinds of null model permutations the authors conducted. Here the authors state that association indices must be measured against a null model. However given the dyadic non-independence of social interaction, all social measures should be modeled against random expectations. It appears later that they may have correctly compared models of all behavior types to null models - line 225. Please clarify.

Paragraph start line 164: I appreciate starting the statistical analysis section with an outline of the number of models to keep track of.

Lines 176 – 177: Clarify here over what time frame annual rank was calculated. Currently it sounds like it was measured on a single day, August 31.

Lines 188 – 195: I am unclear regarding why measures such as sex ratio, number of partners available, and group size are to be included in the repeatability model, when association permutation arguable already controlled for them.

Line 221 – I suggest using consistent terminology and choosing either “association” or “gregariousness” to use throughout. Both can be used when introducing the meaning of association. Also line 260.

Line 233 – Suggest also citing Nakagawa and Schielzeth et al 2010 Biological Reviews.

Table 1 – Good, clear layout of results.

Fig 2 & 3 – Please clarify: the x axis represents how consistent an individual was in its behavior over time, and red indicates how much that individual interacted relative to community average. Is it a coincidence then that all individuals’ that interact less than the population mean are also low on repeatability?

General comment: Does individual level repeatability correlate w years/days observed? This would not be damning if so, but could highlight a limitation of the approach.

Line 310: Add comma after “and”.

Line 313: What are the many former studies? Cite.

Line 318 - 326: This current framing of significance is weak. Humans are the primary subjects of all personality research. It is not surprising that their closest evolutionary relatives also show

repeatable behavior. What do differences in repeatability on annual vs daily scales and between males and females mean about chimpanzee social strategies? I suggest the authors get straight to this meaty interpretation, particularly in paragraphs starting lines 327, 332, 345 & 356. Each of these paragraphs currently leads like a summary of statistical results. I suggest leading with a description of differential power structures and seasonality in males and females, and then tying them to the results in terms of how they would shape within-individual variation in aggressive/friendly behavior.

Line 324: Thompson Cords 2018 Ecology and Evolution also calculate repeatability in grooming in adult female blue monkeys.

Line 355: This idea about constrained preferences and its contrast with flexible choice sounds interesting. Please develop it further and introduce it earlier on.

Paragraph starting line 367: It's unclear what this paragraph is trying to achieve. Is it setting up a future study? Or is it tying the findings into the original discussion of the social niche hypothesis in the introduction? Some kind of return to and evaluation of that original hypothesis would be valuable. Your introduction seemed to lay a promising groundwork for a comparison between social tendencies arising from social niche specialization and/or being temporary life stage strategies. Could you speak to one or both ideas more and what your results mean in relation to them?

Line 338: Replace "generated" with "characterized".

Review form: Reviewer 3

Is the manuscript scientifically sound in its present form?

Yes

Are the interpretations and conclusions justified by the results?

Yes

Is the language acceptable?

Yes

Do you have any ethical concerns with this paper?

No

Have you any concerns about statistical analyses in this paper?

No

Recommendation?

Major revision is needed (please make suggestions in comments)

Comments to the Author(s)

This study investigates the long term repeatability in social behaviors in wild chimpanzees. This repeatability could suggest stable social phenotypes. The study resulted in a dataset of many individuals and data covering more than 20 years (however, the individual mean is only 6 years and the maximum 15 years). During a short period, social bonds adapt because of the instability of dominance hierarchies, fluctuation resources availability, individual states etc., adaptation needs flexibility. Besides, there is a consistent individual difference in social behavior in a various range of taxa in a long period of time. There is a stable tendency in interactions. The degree of repeatability is linked to genetic and stable adaptations due to the experiences of the individual

during his development. The social strategies of each individual are based on their own characteristics. Long term study are interesting to show individual difference independent from a special step in life cycle. It permits to see if the social behavior is reproducible over a long period of time.

I read this manuscript favorably and believe this is worth publication, but have also some concerns.

My biggest concern is the time frame. The terms of analyzed data from the 45 individuals varied from 3 years to 15 years (mean = 6 years). I could not understand why this is so short considering their long life span (>50 years) and the studies' length (20 years). 6 years are not enough to see the chimpanzees' life-long stability. The description of the abstract "Using data spanning over 20 years, we demonstrate that multiple social behaviours are repeatable over the long-term in wild chimpanzees" is misleading.

Grooming is an important component of social bond formation. The contingent grooming choice is based on a wide range of parameters such as audience, partner rank or context, as reconciliation after aggression for example. Grooming is used to reach social goals as dominance rank and formation of social bonds which have a huge influence on fitness. For the grooming they extracted the time focal individuals spent grooming adult partners. They specified "we focused on grooming given to others rather than overall time grooming, i.e. including grooming received, as this would reflect a tendency to attract grooming partners rather than an individual tendency to groom". This explanation is not very clear to me. Does it mean that they did not use mutual grooming? Yet mutual grooming seems important to measure the strength of social bonds. This point may need further clarifications.

Social trends seem to be important. Understanding how some individuals become more aggressive, affiliated or gregarious, than others, requires further empirical explorations. However, in the discussion they only mention the immature period during which young chimpanzees are systematically linked to their mother. Much of the discussion is focused on this specific point. It might have been interesting to raise other factors. The social niche hypothesis suggests that coherent individual behavioral differences occur due to the specialization of the niche to improve intra-species and/or intra-group competition for resources. However this theory is mentioned only briefly without any explication. I think this could have been deepened.

Decision letter (RSOS-200454.R0)

Dear Dr Tkaczynski,

The editors assigned to your paper ("Long-term repeatability in social behaviours suggests stable social phenotypes in wild chimpanzees") have now received comments from reviewers. We would like you to revise your paper in accordance with the referee and Associate Editor suggestions which can be found below (not including confidential reports to the Editor). Please note this decision does not guarantee eventual acceptance.

Please submit a copy of your revised paper before 30-May-2020. Please note that the revision deadline will expire at 00.00am on this date. If we do not hear from you within this time then it will be assumed that the paper has been withdrawn. In exceptional circumstances, extensions may be possible if agreed with the Editorial Office in advance. We do not allow multiple rounds

of revision so we urge you to make every effort to fully address all of the comments at this stage. If deemed necessary by the Editors, your manuscript will be sent back to one or more of the original reviewers for assessment. If the original reviewers are not available, we may invite new reviewers.

- Data accessibility

<http://datadryad.org/submit?journalID=RSOS&manu=RSOS-200454>

- Competing interests

- Authors' contributions

- Acknowledgements

- Funding statement

on behalf of Dr Atsushi Iriki (Associate Editor) and Pete Smith (Subject Editor)
openscience@royalsociety.org

Comments to Author:

Reviewers' Comments to Author:

Reviewer: 1

Comments to the Author(s)

See attached file.

Reviewer: 2

Comments to the Author(s)

The authors examine consistent between-individual differences in agonistic and affiliative social behavior in wild chimpanzees, using a dense longitudinal sample of individuals living in 3 distinct communities in the Tai forest of Cote D'Ivoire. The authors specifically look at repeatability in daily vs annual levels of aggression, party association, and grooming given. The authors find that individual chimps are generally consistent in their preferences to associate in parties of a particular size and the amount of time they spend in association, though on an annual level males were less consistent than females. Males were also less consistent than females in their tendency to give aggression and grooming.

Recent examinations of consistent inter-individual differences, such as this study, provide important insight into behavioral strategies that deviate from population averages, and that were historically considered noise surrounding species-typical behavior. They also lend important insight into the broader evolutionary and ecological significance of what are usually termed "personalities" in humans. What I believe would strengthen this paper is to contextualize the inter-individual differences better in terms of social strategies that are relevant to chimpanzee social life. I flesh out specific areas for improvement in the following line-by-line comments.

Line 36-38: The length of previous collection time periods alone does not necessarily warrant further research in consistent individual differences. There is quite a lot of emphasis on the size of data set in this study as a measure of its significance. While the data set is impressive for a long-lived animal and certainly hard-won, many species examined for repeatable behavior have been observed over similarly long periods relative to their lifespans (many insects, birds, and fish, e.g. 3 consecutive flocking seasons in great tits that live to 13 years at maximum, Aplin et al. 2015 *Animal Behavior*). Observation years per individual in this study ranged 6 - 12 years in this study, for an animal that lives perhaps up to 65. Observations in chimps are perhaps more consistent and even over each year - how might this make for particularly reliable and robust estimations of repeatability?

Intro paragraph starting line 47: Again, what is the significance of repeatable behavior? Currently this paragraph highlights various internal and external stimuli that can cause behavior to vary. While this is true, an interesting aim is to determine whether behavior, which has elsewhere been determined to be adaptive (e.g. affiliation, coalitionary aggression), is flexible to the moment or representative of a more constant trait.

Line 91: I suggesting strengthening the justification for this analysis in chimpanzees. The authors seem to present 2 hypotheses in the introduction, either personality is canalized by early environmental density dependent conditions (social niche), or behavior varies by life history stage and/or socio-ecological condition. If these are two hypotheses the authors wish to test, how are fission fusion dynamics relevant?

Line 145: Clarify, does association mean spatial association/party membership excluding time grooming, or do these 2 measures overlap?

Line 153: I am confused as to the kinds of null model permutations the authors conducted. Here the authors state that association indices must be measured against a null model. However given the dyadic non-independence of social interaction, all social measures should be modeled against random expectations. It appears later that they may have correctly compared models of all behavior types to null models - line 225. Please clarify.

Paragraph start line 164: I appreciate starting the statistical analysis section with an outline of the number of models to keep track of.

Lines 176 - 177: Clarify here over what time frame annual rank was calculated. Currently it sounds like it was measured on a single day, August 31.

Lines 188 - 195: I am unclear regarding why measures such as sex ratio, number of partners available, and group size are to be included in the repeatability model, when association permutation arguable already controlled for them.

Line 221 - I suggest using consistent terminology and choosing either "association" or "gregariousness" to use throughout. Both can be used when introducing the meaning of association. Also line 260.

Line 233 - Suggest also citing Nakagawa and Schielzeth et al 2010 *Biological Reviews*.

Table 1 - Good, clear layout of results.

Fig 2 & 3 - Please clarify: the x axis represents how consistent an individual was in its behavior over time, and red indicates how much that individual interacted relative to community average. Is it a coincidence then that all individuals' that interact less than the population mean are also low on repeatability?

General comment: Does individual level repeatability correlate w years/days observed? This would not be damning if so, but could highlight a limitation of the approach.

Line 310: Add comma after “and”.

Line 313: What are the many former studies? Cite.

Line 318 - 326: This current framing of significance is weak. Humans are the primary subjects of all personality research. It is not surprising that their closest evolutionary relatives also show repeatable behavior. What do differences in repeatability on annual vs daily scales and between males and females mean about chimpanzee social strategies? I suggest the authors get straight to this meaty interpretation, particularly in paragraphs starting lines 327, 332, 345 & 356. Each of these paragraphs currently leads like a summary of statistical results. I suggest leading with a description of differential power structures and seasonality in males and females, and then tying them to the results in terms of how they would shape within-individual variation in aggressive/friendly behavior.

Line 324: Thompson Cords 2018 Ecology and Evolution also calculate repeatability in grooming in adult female blue monkeys.

Line 355: This idea about constrained preferences and its contrast with flexible choice sounds interesting. Please develop it further and introduce it earlier on.

Paragraph starting line 367: It's unclear what this paragraph is trying to achieve. Is it setting up a future study? Or is it tying the findings into the original discussion of the social niche hypothesis in the introduction? Some kind of return to and evaluation of that original hypothesis would be valuable. Your introduction seemed to lay a promising groundwork for a comparison between social tendencies arising from social niche specialization and/or being temporary life stage strategies. Could you speak to one or both ideas more and what your results mean in relation to them?

Line 338: Replace “generated” with “characterized”.

Reviewer: 3

Comments to the Author(s)

This study investigates the long term repeatability in social behaviors in wild chimpanzees. This repeatability could suggest stable social phenotypes. The study resulted in a dataset of many individuals and data covering more than 20 years (however, the individual mean is only 6 years and the maximum 15 years). During a short period, social bonds adapt because of the instability of dominance hierarchies, fluctuation resources availability, individual states etc., adaptation needs flexibility. Besides, there is a consistent individual difference in social behavior in a various range of taxa in a long period of time. There is a stable tendency in interactions. The degree of repeatability is linked to genetic and stable adaptations due to the experiences of the individual during his development. The social strategies of each individual are based on their own characteristics. Long term study are interesting to show individual difference independent from a special step in life cycle. It permits to see if the social behavior in reproducible over a long period of time.

I read this manuscript favorably and believe this is worth publication, but have also some concerns.

My biggest concern is the time frame. The terms of analyzed data from the 45 individuals varied from 3 years to 15 years (mean = 6 years). I could not understand why this is so short considering their long life span (>50 years) and the studies' length (20 years). 6 years are not enough to see the

chimpanzees' life-long stability. The description of the abstract "Using data spanning over 20 years, we demonstrate that multiple social behaviours are repeatable over the long-term in wild chimpanzees" is misleading.

Grooming is an important component of social bond formation. The contingent grooming choice is based on a wide range of parameters such as audience, partner rank or context, as reconciliation after aggression for example. Grooming is used to reach social goals as dominance rank and formation of social bonds which have a huge influence on fitness. For the grooming they extracted the time focal individuals spent grooming adult partners. They specified "we focused on grooming given to others rather than overall time grooming, i.e. including grooming received, as this would reflect a tendency to attract grooming partners rather than an individual tendency to groom". This explanation is not very clear to me. Does it mean that they did not use mutual grooming? Yet mutual grooming seems important to measure the strength of social bonds. This point may need further clarifications.

Social trends seem to be important. Understanding how some individuals become more aggressive, affiliated or gregarious, than others, requires further empirical explorations. However, in the discussion they only mention the immature period during which young chimpanzees are systematically linked to their mother. Much of the discussion is focused on this specific point. It might have been interesting to raise other factors. The social niche hypothesis suggests that coherent individual behavioral differences occur due to the specialization of the niche to improve intra-species and/or intra-group competition for resources. However this theory is mentioned only briefly without any explication. I think this could have been deepened.

Author's Response to Decision Letter for (RSOS-200454.R0)

See Appendix B.

RSOS-200454.R1 (Revision)

Review form: Reviewer 1

Is the manuscript scientifically sound in its present form?

No

Are the interpretations and conclusions justified by the results?

Yes

Is the language acceptable?

No

Do you have any ethical concerns with this paper?

No

Have you any concerns about statistical analyses in this paper?

No

Recommendation?

Accept with minor revision (please list in comments)

Comments to the Author(s)

This revised manuscript is a marked improvement over the previous version. The introduction gives a clearer lead-in to the study, and it is much easier for a reader to follow the methods and arguments. However, I still think the authors can improve the organization, especially of the Discussion. It would help a lot if they followed the roll out of predictions at the end of the Introduction in organizing the Discussion, i.e. use the same predictions in the same order in both sections of the paper. It seems there are three natural sections: comparisons of Repeatability to other reports/taxa/behaviors, comparison among the three types of behavior examined here, and comparisons between the sexes (for each of the three behaviors). Comparisons between the two data aggregation scales might be a fourth. Set up a parallel structure when discussing these comparisons so that the Introduction and Discussion mirror each other in terms of organization/structure (at least, for discussing these particular results). Make sure the Discussion states explicitly whether expectations were met or not.

I also found the Discussion a little longer than I think it needs to be, especially the very last section. Perhaps this can be reduced a little. These are interesting questions but the data really cannot address them.

I am still concerned about two aspects of how the dataset was put together. For the yearly data set, why take one single daily value (Aug 31) to represent an entire year (instead of averaging across days of the year? I don't see how it can possibly be true that the value on one date is a better representation of the "whole year" than some kind of average (median, mean). The authors have also not justified why it is ok to accept follows that are only 3 hours long as representative of a "dawn to dusk" follow: does a follow of this length represent an accurate assessment of the behavioral variables used in the analysis, especially when data are collated on a daily basis? The authors need to make their case here, or possibly swap out some measurements. The latter would be a bigger deal, I realize, as reanalysis would be required.

Line by line comments: most relate to expressing things correctly or more effectively. A few are other sorts of questions.

58: I would say "often" rather than "generally". So few species have been examined, it's hard to know if it's really general.

60: "evidenced repeatability" is not correct English (to evidence is not a verb)

63: "vary" would be better than "fluctuate" (see also 115 for noun form, and several other cases in the manuscript - I think VARY sounds better in ALL of them)

78: typically one does not use "etc" in formal writing

95: Does everything have to be a "model" species? Personally I don't think any primate is a model species: that term is usually refers to the lab mouse/rat and their ilk. Chimpanzees and primates are seldom model species because they are way too hard to work with. How about "useful" or "appropriate" or "interesting"? I think it's a plus that this paper is NOT about a model species!

99: do you mean "life history STAGE" here?

107ff: it is convention to use past tense in writing about one's own study... many changes needed

118: "affect", rather than "impact on"

120: word missing? to BE flexible?

122: consider giving SOME idea of what “longer term” means here – years?

125: replace “the composition of bystanders” (unclear) with “which bystanders are nearby”; where you refer to “rank differences with available partners”, presumably you mean differences in the subject’s rank relative to different partners, so just refer to partner rank, not rank differences?

132: “our other behaviors of interest” – a reader doesn’t know what you mean here... perhaps add (see below)? Also, no reason to use the possessive.

136: “their” technically refers to “ranks” (the last plural noun) ... rewording needed here

144-146: this sentence needs some rewriting – better not to use future tense, and in general it’s confusing, possibly some words missing

149: adjust –explain with a brief reference that the adjustment is carried out by choosing which sized party to join or remain with

155, 501: extant?? are you trying to say the predation pressure is high?

159: which “studies to date”? all of them? More importantly, it’s not clear why you expect this. Did other studies include more variable life stages (not only adults)? You go on to describe how many things change both for males and females during adulthood, so this text seems to argue AGAINST the idea that limiting the study to adults should lead to an expectation of low repeatability. I was left a bit baffled.

175: here you reference “nest to nest” focal follows, but then later you say the follows had to last (only) 3 hrs... this seems inconsistent. Did all the short (not full day) follows start at a night nest in the morning? Are there diurnal rhythms of activity, especially the behaviors you examined, in chimps and if samples were biased by time of day (all started early, fewer afternoons represented), isn’t that a concern? Explain for the reader.

Table 1: For the daily analyses, there appears to be no information about how the individual days were spread over time. Can you please provide this information or clarify?

238: “other individuals” is still only adults, right?

241: what kinds of behavior were included as aggressive?

254: omit apostrophe

261: by CHANCE, not by random

263: what do you mean by “subsequent parties that originally had the same party membership” – what is a subsequent party? Not quite following here.

278: why would sex ratio not be calculated as the average across all days of the year?

283: in, not into. For yearly group size, when do you measure this? Presumably group size may change over the course of a year, so is it a time-averaged group size across the days of the year?

284: based on, not based

292: even those who were not yet adult at the beginning of habituation had estimated ages, right? If you don’t KNOW their date of birth, you must be estimating their age: you just have more to

go on in these cases, as you witness them growing/changing more than if they were already adult at the start of habituation.

296: why do you assign age at the beginning of the year instead of the year's midpoint? Why is age assessed at the beginning of the year whereas sex ratio (and rank?) is assessed at the end of the year?

300: A reader should not have to read additional papers to understand what you did here. Please provide a little more information: what kind of modification was made, were all pant grunt interactions between adults only, both sexes together? Finally, if rank changes, especially for males, I think using a rank on ONE day of the year needs strong justification: why not take the average Elo rating for the individual across the year?

314-315: are these separate variables or different levels of a single categorical variable? During a year, a female might have a newborn AND unweaned offspring (the newborn grows)... which takes priority then?

336: you have not yet described cosine and sine functions?

341: so does this rate vary, in principle, from 0 to 1? Clarify.

348: unique individuals? or average # individuals in the party/parties?

365: you mean EXAMINED, not ESTABLISHED, I think

382: instead of "there were few differences" say "differences were minimal"

Table 2: The legend should explain the column headers more (perhaps move some info from the text to the table, or repeat it briefly)

Fig. 1: maybe jitter the symbols for grooming, or indicate in legend that male and female values coincide and are superpositioned. Figure legend: "delineated" is not the right verb. I think this legend could be written more clearly - explain what R2 is, briefly.

Fig 2-3: legend includes phrase "given all fixed and random effects" - this sounds a bit odd. Also avoid having the word "given" twice in a sentence.

439: I believe one should not use behaviors as a plural noun.. maybe "types of behavior" or "behavior types"? This is an issue in multiple places later as well.

458: allows ONE (add the word "one")

472: differences EXTEND (not extends)

472-473: why do you say they should influence fitness more? Don't require the reader to examine 3 additional papers. Are you speaking only about chimpanzees or is this intended as a broader statement?

479: this statement directly contradicts 507

488: what is meant by "physically compete"? You mean directly, i.e. aggressively?

489-490: this sentence needs work, seems like words are missing or the lack of parallel construction just makes it hard to follow

501: to, or according to?

523: life history STRATEGIES? What do you mean by this?

543: this is a dangling modifier: the chimpanzees didn't reveal

549: years ARE required, not IS required

551: data ALLOW, not allows

554: This what?

Review form: Reviewer 2

Is the manuscript scientifically sound in its present form?

Yes

Are the interpretations and conclusions justified by the results?

Yes

Is the language acceptable?

Yes

Do you have any ethical concerns with this paper?

No

Have you any concerns about statistical analyses in this paper?

No

Recommendation?

Accept with minor revision (please list in comments)

Comments to the Author(s)

The authors have improved their introduction greatly, with a much better focus on the potential significance of repeatable behavior in chimpanzees. The study is set up in the last paragraph of the introduction with clear hypotheses and predictions. In Methods, the reason for permutation methods for significance testing is clear now, as association strength is a social network measure. In Discussion, the authors take the appropriate room to discuss different time frames for aggregation.

The following are my remaining concerns by line number:

Line 154 - Guide the reader as to what "more gregarious" means - each individual at Tai spends a larger amount of time, on average, in a social party than chimpanzees at other sites do?

Lines 243 - 269. Still unclear whether spatial association excludes time spent grooming.

Line 440-442 - The opening of your discussion would be stronger by stating the significance or biological meaning of your results coming near the meta-analytical $R = 0.32$ (note, I believe this should be "R" and not "R²"). Your last sentence of this paragraph seems to be the big takeaway and I suggest moving it towards the beginning of the paragraph.

Line 468 - I recommend starting this paragraph by stating the new idea to discuss, rather than reiterating your result.

Line 517 – I suggest just calling their social groups “fission-fusion” instead of “complex”.

Line 517 I suggest that at the beginning of the section “Causes and Consequences of Repeatable Social Behaviour” you bring the reader back to the last paragraph of your intro, where you had competing hypotheses. Highlight that you've found evidence more in favor of one than the other, e.g. "Given repeatability in social behavior, independent of factors related to environmental and physiological conditions, we find preliminary support for early life canalization of social phenotypes..." Do acknowledge that there may be other variables that you didn't control for that could constrain social behavior during adulthood. This acknowledgment of limitations would more reasonably present early life canalization as not the definitive cause of repeatability but one worthy of further exploration.

Paragraphs starting line 531 and 547

While I appreciate that the authors are probably setting up their future study on social niche specialization, the 2 paragraphs dedicated to this as *the* origin of repeatability (lines 531-556) can be abbreviated possibly to one paragraph. Currently, it seems like too much space and thought is dedicated to a topic that is actually somewhat peripheral to the paper.

Decision letter (RSOS-200454.R1)

Dear Dr Tkaczynski:

On behalf of the Editors, I am pleased to inform you that your Manuscript RSOS-200454.R1 entitled "Long-term repeatability in social behaviours suggests stable social phenotypes in wild chimpanzees" has been accepted for publication in Royal Society Open Science subject to minor revision in accordance with the referee suggestions. Please find the referees' comments at the end of this email.

The reviewers and Subject Editor have recommended publication, but also suggest some minor revisions to your manuscript. Therefore, I invite you to respond to the comments and revise your manuscript.

- Ethics statement

- Data accessibility

If you wish to submit your supporting data or code to Dryad (<http://datadryad.org/>), or modify your current submission to dryad, please use the following link:
<http://datadryad.org/submit?journalID=RSOS&manu=RSOS-200454.R1>

- Competing interests

- Authors' contributions

- Acknowledgements

- Funding statement

Because the schedule for publication is very tight, it is a condition of publication that you submit the revised version of your manuscript before 03-Jul-2020. Please note that the revision deadline will expire at 00.00am on this date. If you do not think you will be able to meet this date please let me know immediately.

on behalf of Dr Atsushi Iriki (Associate Editor) and Pete Smith (Subject Editor)
openscience@royalsociety.org

Reviewer comments to Author:
 Reviewer: 1

Comments to the Author(s)

This revised manuscript is a marked improvement over the previous version. The introduction gives a clearer lead-in to the study, and it is much easier for a reader to follow the methods and arguments. However, I still think the authors can improve the organization, especially of the Discussion. It would help a lot if they followed the roll out of predictions at the end of the Introduction in organizing the Discussion, i.e. use the same predictions in the same order in both sections of the paper. It seems there are three natural sections: comparisons of Repeatability to other reports/taxa/behaviors, comparison among the three types of behavior examined here, and comparisons between the sexes (for each of the three behaviors). Comparisons between the two data aggregation scales might be a fourth. Set up a parallel structure when discussing these comparisons so that the Introduction and Discussion mirror each other in terms of organization/structure (at least, for discussing these particular results). Make sure the Discussion states explicitly whether expectations were met or not.

I also found the Discussion a little longer than I think it needs to be, especially the very last section. Perhaps this can be reduced a little. These are interesting questions but the data really cannot address them.

I am still concerned about two aspects of how the dataset was put together. For the yearly data set, why take one single daily value (Aug 31) to represent an entire year (instead of averaging across days of the year? I don't see how it can possibly be true that the value on one date is a better representation of the "whole year" than some kind of average (median, mean). The authors have also not justified why it is ok to accept follows that are only 3 hours long as representative of a "dawn to dusk" follow: does a follow of this length represent an accurate assessment of the behavioral variables used in the analysis, especially when data are collated on a daily basis? The authors need to make their case here, or possibly swap out some measurements. The latter would be a bigger deal, I realize, as reanalysis would be required.

Line by line comments: most relate to expressing things correctly or more effectively. A few are other sorts of questions.

58: I would say "often" rather than "generally". So few species have been examined, it's hard to know if it's really general.

60: "evidenced repeatability" is not correct English (to evidence is not a verb)

63: "vary" would be better than "fluctuate" (see also 115 for noun form, and several other cases in the manuscript – I think VARY sounds better in ALL of them)

78: typically one does not use "etc" in formal writing

95: Does everything have to be a "model" species? Personally I don't think any primate is a model species: that term usually refers to the lab mouse/rat and their ilk. Chimpanzees and primates are seldom model species because they are way too hard to work with. How about "useful" or "appropriate" or "interesting"? I think it's a plus that this paper is NOT about a model species!

99: do you mean "life history STAGE" here?

107ff: it is convention to use past tense in writing about one's own study... many changes needed

118: "affect", rather than "impact on"

120: word missing? to BE flexible?

122: consider giving SOME idea of what "longer term" means here – years?

125: replace "the composition of bystanders" (unclear) with "which bystanders are nearby"; where you refer to "rank differences with available partners", presumably you mean differences in the subject's rank relative to different partners, so just refer to partner rank, not rank differences?

132: "our other behaviors of interest" – a reader doesn't know what you mean here... perhaps add (see below)? Also, no reason to use the possessive.

136: "their" technically refers to "ranks" (the last plural noun) ... rewording needed here

144-146: this sentence needs some rewriting – better not to use future tense, and in general it's confusing, possibly some words missing

149: adjust –explain with a brief reference that the adjustment is carried out by choosing which sized party to join or remain with

155, 501: extant?? are you trying to say the predation pressure is high?

159: which “studies to date”? all of them? More importantly, it’s not clear why you expect this. Did other studies include more variable life stages (not only adults)? You go on to describe how many things change both for males and females during adulthood, so this text seems to argue AGAINST the idea that limiting the study to adults should lead to an expectation of low repeatability. I was left a bit baffled.

175: here you reference “nest to nest” focal follows, but then later you say the follows had to last (only) 3 hrs... this seems inconsistent. Did all the short (not full day) follows start at a night nest in the morning? Are there diurnal rhythms of activity, especially the behaviors you examined, in chimps and if samples were biased by time of day (all started early, fewer afternoons represented), isn’t that a concern? Explain for the reader.

Table 1: For the daily analyses, there appears to be no information about how the individual days were spread over time. Can you please provide this information or clarify?

238: “other individuals” is still only adults, right?

241: what kinds of behavior were included as aggressive?

254: omit apostrophe

261: by CHANCE, not by random

263: what do you mean by “subsequent parties that originally had the same party membership” – what is a subsequent party? Not quite following here.

278: why would sex ratio not be calculated as the average across all days of the year?

283: in, not into. For yearly group size, when do you measure this? Presumably group size may change over the course of a year, so is it a time-averaged group size across the days of the year?

284: based on, not based

292: even those who were not yet adult at the beginning of habituation had estimated ages, right? If you don’t KNOW their date of birth, you must be estimating their age: you just have more to go on in these cases, as you witness them growing/changing more than if they were already adult at the start of habituation.

296: why do you assign age at the beginning of the year instead of the year’s midpoint? Why is age assessed at the beginning of the year whereas sex ratio (and rank?) is assessed at the end of the year?

300: A reader should not have to read additional papers to understand what you did here. Please provide a little more information: what kind of modification was made, were all pant grunt interactions between adults only, both sexes together? Finally, if rank changes, especially for males, I think using a rank on ONE day of the year needs strong justification: why not take the average Elo rating for the individual across the year?

314-315: are these separate variables or different levels of a single categorical variable? During a year, a female might have a newborn AND unweaned offspring (the newborn grows)... which takes priority then?

336: you have not yet described cosine and sine functions?

341: so does this rate vary, in principle, from 0 to 1? Clarify.

348: unique individuals? or average # individuals in the party/parties?

365: you mean EXAMINED, not ESTABLISHED, I think

382: instead of "there were few differences" say "differences were minimal"

Table 2: The legend should explain the column headers more (perhaps move some info from the text to the table, or repeat it briefly)

Fig. 1: maybe jitter the symbols for grooming, or indicate in legend that male and female values coincide and are superpositioned. Figure legend: "delineated" is not the right verb. I think this legend could be written more clearly - explain what R2 is, briefly.

Fig 2-3: legend includes phrase "given all fixed and random effects" - this sounds a bit odd. Also avoid having the word "given" twice in a sentence.

439: I believe one should not use behaviors as a plural noun.. maybe "types of behavior" or "behavior types"? This is an issue in multiple places later as well.

458: allows ONE (add the word "one")

472: differences EXTEND (not extends)

472-473: why do you say they should influence fitness more? Don't require the reader to examine 3 additional papers. Are you speaking only about chimpanzees or is this intended as a broader statement?

479: this statement directly contradicts 507

488: what is meant by "physically compete"? You mean directly, i.e. aggressively?

489-490: this sentence needs work, seems like words are missing or the lack of parallel construction just makes it hard to follow

501: to, or according to?

523: life history STRATEGIES? What do you mean by this?

543: this is a dangling modifier: the chimpanzees didn't reveal

549: years ARE required, not IS required

551: data ALLOW, not allows

554: This what?

Reviewer: 2

Comments to the Author(s)

The authors have improved their introduction greatly, with a much better focus on the potential significance of repeatable behavior in chimpanzees. The study is set up in the last paragraph of

the introduction with clear hypotheses and predictions. In Methods, the reason for permutation methods for significance testing is clear now, as association strength is a social network measure. In Discussion, the authors take the appropriate room to discuss different time frames for aggregation.

The following are my remaining concerns by line number:

Line 154 – Guide the reader as to what “more gregarious” means – each individual at Tai spends a larger amount of time, on average, in a social party than chimpanzees at other sites do?

Lines 243 – 269. Still unclear whether spatial association excludes time spent grooming.

Line 440-442 – The opening of your discussion would be stronger by stating the significance or biological meaning of your results coming near the meta-analytical $R = 0.32$ (note, I believe this should be “R” and not “R2”). Your last sentence of this paragraph seems to be the big takeaway and I suggest moving it towards the beginning of the paragraph.

Line 468 – I recommend starting this paragraph by stating the new idea to discuss, rather than reiterating your result.

Line 517 – I suggest just calling their social groups “fission-fusion” instead of “complex”.

Line 517 I suggest that at the beginning of the section “Causes and Consequences of Repeatable Social Behaviour” you bring the reader back to the last paragraph of your intro, where you had competing hypotheses. Highlight that you've found evidence more in favor of one than the other, e.g. "Given repeatability in social behavior, independent of factors related to environmental and physiological conditions, we find preliminary support for early life canalization of social phenotypes..." Do acknowledge that there may be other variables that you didn't control for that could constrain social behavior during adulthood. This acknowledgment of limitations would more reasonably present early life canalization as not the definitive cause of repeatability but one worthy of further exploration.

Paragraphs starting line 531 and 547

While I appreciate that the authors are probably setting up their future study on social niche specialization, the 2 paragraphs dedicated to this as *the* origin of repeatability (lines 531-556) can be abbreviated possibly to one paragraph. Currently, it seems like too much space and thought is dedicated to a topic that is actually somewhat peripheral to the paper.

Author's Response to Decision Letter for (RSOS-200454.R1)

See Appendix C.

Decision letter (RSOS-200454.R2)

Dear Dr Tkaczynski,

It is a pleasure to accept your manuscript entitled "Long-term repeatability in social behaviour suggests stable social phenotypes in wild chimpanzees" in its current form for publication in Royal Society Open Science.

Best regards,

on behalf of Dr Atsushi Iriki (Associate Editor) and Pete Smith (Subject Editor)
openscience@royalsociety.org

Appendix A

This study compares repeatability measurements for three aspects of social behavior (rates of grooming (duration), aggression (count) and association/gregariousness (index expressing deviation from random expectation)) in male and female members of a wild chimpanzee population studied individually for periods up to several years. The stated aim is to use data from a long-term study to assess intraindividual variation in behavior over time, as well as interindividual differences that cannot be explained by time-varying situational or individual attributes, such as group size, age and rank. In the Introduction, the authors articulate several predictions based on their understanding of behavioral variation over the lifespan in the study species. Specifically, they expect (1) greater intraindividual variation in rates of aggression for males than females, (2) overall low repeatability for rates of grooming given (this prediction would benefit from some comparative – low with relative to what?), and (3) low repeatability for association (again, relative to what?). Later (in Methods), it becomes clear that they also bin their data for the analysis in two ways, by day and by year, although the predictions articulated in the Introduction do not refer to these two bin sizes and one has to read the entire paper to understand the relevance of these alternative data-collation procedures. This comparison is interesting, however, and does have implications (not developed, but briefly mentioned in the Discussion) for interpreting analyses of individual differences comparatively. Differences in repeatability for the three behaviors, between sexes and between the two time-frames analyzed, are interpreted in a reasonable way.

Overall, this was an interesting paper based on rare longitudinal data in a highly social and very long-lived species. However, the presentation is quite complex (many comparisons, complex analysis), and the manuscript was a very difficult read. It would benefit from much tighter organization, with goals articulated more clearly in the Introduction, and retention of the organizational roll-out from the Introduction in the Methods (as much as possible) and in the Results and Discussion. Avoid using varying terminology. Be very careful with word choice. Specific comments below may help.

Also, the data set itself should be better described in the text. Follows of chimpanzees under natural conditions are, I believe, always somewhat serendipitous, as these primates do not occur in stable associations. The authors do not describe how focal subjects were chosen (how randomly?), nor how the follows of individual subjects were distributed over time, which seems important if one of the motivations of the study is accurate assessment of behavior over time through the use of a long-term data set. Information on the data set is available in a supplementary table, but I would recommend summarizing key information in the text, relating especially to variation in the amount of data per individual, both in terms of regularity of observations and total time span over which observations occurred.

Introduction

It would benefit communication if there was some reorganization of the introduction so that a reader knows earlier what this study will be about. The first several paragraphs are not particularly focused or well organized or written. I even think there could be more compelling “hooks:” you start out with social bonds and connectivity, but then at the end of the second paragraph we’re onto cooperativeness (which might be related) but also gregariousness (which is different from bonding) and aggressiveness (which seems like

something else altogether). You don't get to repeatability until the third paragraph. And not until after this, as you lay out predictions, does a reader realize that your main aim is to compare repeatability for different sexes and behavior types, and for data organized on a daily vs yearly basis. Make the conclusions one can draw from those comparisons the focus of the paper, right at the beginning.

The introduction initially seems to describe the motivation of the study in terms of the necessity of covering enough of the organism's lifespan that one will avoid misinterpreting as persistent individual differences what might just be differences reflecting age, rank or demographic variables. This is a valid methodological point, but the text could do a better job making it convincingly. It would help to present (at least review) more specific information about the limitations of studies to date and highlighting any reports that compare a shorter vs. longer-duration data sets, and how conclusions differ. I also think the methodological limitations of prior work is a secondary point (and therefore should be placed later than it is in the Introduction) to the main aim here, which is comparing sexes and behavior types (and time frames) in terms of the repeatability they exhibit. To achieve this aim, of course one wants to measure repeatability well, and that's where having a long-enough time span is crucial. This said, it is not entirely clear that your data are an improvement on other data: data from East group come from only a 4 year period, which is arguably not a very big chunk of a chimpanzee's lifespan. Again, more description of the data set will help to make your case.

Lines 58-59: State earlier what kind of social behavior differences you are referring to here. This sentence is really much more related to the subject of your report than the paragraph before, and a reader wants you to dig in here. The last sentence of the paragraph provides some details, but without references.

Line 70-71 is a little confusing as you're trying to explain "consistent individual differences" (line 68) and yet you are invoking features of the individual that *change* over time (body size, health, dominance possibly).

Line 73 ff: it seems you are saying that one can, if data are limited, misinterpret APPARENT individual differences which are actually related changeable characteristics to age, rank, etc as being "true" or persistent individual differences. But if you write that the differences are "due to" these (changeable) features of the individual, then you appear to be contradicting yourself.

84: If the pitfalls of interpreting apparent individual differences from limited data are motivating your study, it would be useful to review this issue more: what kinds of time frames are common in the literature, have studies with longer durations reached different conclusions than those with shorter durations (ideally of the same organism)? How well have factors that might change over time been controlled in previous analyses? After reading the whole paper, however, it seemed that perhaps I misinterpreted what motivated your study – see comments at the beginning of this review. Even so, it would be helpful to say a bit more about adequate sample sizes for assessing repeatability.

94: does “social organization” really fluctuate? What exactly do you mean by social organization here?

99ff: Rank and “aggressive tendencies” seem to be viewed as the same thing here. I’m not entirely sure what “aggressive tendencies” are (later, 103, also referred to as “aggression”), but it need not be the case that the rate (per unit time) of aggressive interactions is correlated with rank in animal societies. If they are correlated in chimpanzee societies, that point should be made explicitly. And using precise language is very important.

105: shouldn’t you start a new paragraph when you describe grooming? And when you say “grooming behavior” please (again) be more specific: time spent grooming? number or diversity of partners? what exactly? Similarly, later you reference “association behavior” but what do you mean specifically? This too should be clear from the get go, to facilitate the reader’s comprehension.

Methods

The Methods section is very complex, and it is a challenge to present the information in a way that others can easily follow. The authors appear to use varying terminology which is not helpful.

124: Authors state that they include as subjects only those individuals who were sampled *regularly*, but more information is needed on the specific criteria: how regularly? How much variation in regularity? The description of data collection provides scant information and although the supplementary document would allow a reader to assess, it would be helpful to provide a little more basic summary in the text of the paper regarding how days and years are distributed over time for the individuals chosen as subjects.

146: clarify what you mean by “cumulative” – is this simply a count of unique adult individuals with whom it associated in a given day?

150: “summing their association” is unclear. What is “their association”?

151: is this an appropriate standardization? If I am following, this would lead you to express the average association strength across all individuals in the subject’s community. But do we expect average values to be similar across communities when they differ in size? If chimpanzees prioritize being in association with a certain number of individuals at a time, for example, then the average association rate per possible partner must be higher in the smaller than in the larger community. Also, how does this standardization relates to the text at 153 ff?

169: estimated with what degree of precision? How many subjects had ages estimated this way? How would uncertainty in ages affect results?

177: so age was extracted for the beginning of the annual interval (171) and rank at the end that interval? This seems inconsistent. Why not compute an average age/rank throughout the year? Same for sex ratio (and is the sex ratio for adults only)?

193: but ARE interaction rates related to group size? This would not *have* to be the case. See comment about line 151 for a similar point.

194-195: The text here is unclear as written, though I can see why for a repeatability analysis you'd want the standardization to be by group.

196-198: This is also unclear. An offset would make sense if one were modelling counts (e.g. for aggression count), but how does it help if one is modeling rates (hourly grooming rate (line 216) and association index)? Would one expect the amount of observation time to influence the proportion of time spent grooming or the association index? Justify.

199: group by year – do you mean “group-year”? Is this the same as “Year within group” referenced on line 210? Data collected from a given group within a given year may not be independent, but then data collected on Aug 30 are probably also not independent of those collected on Sept 2. I am not sure if there may be a better solution here to account for temporal clustering in the data in a series of successive time blocks.

Also, here you say “radians of Julian date” and later you refer to sin and cosin... Please clarify how you account for seasonality and, more generally, be consistent in the terminology.

208: why not take the group size on the same day? Why take a yearly average?

208: why is group ID included as a fixed effect?

216: do you mean logit, not log (Warton and Hui 2011)? Was there zero inflation in the grooming data? Is a Gaussian model appropriate for % of time spent grooming? That is, do transformed values satisfy linear modeling assumptions? Proportions of continuous variables like time are often very tricky as they are bounded by 0 and 1.

In general, have the authors conducted any model diagnostics?

Results

In general, I recommend reporting how many lines of data you had for each model, possibly per subject. That is, how many repeated time units occurred for N subjects?

Because there were no predictions made about analyses based on yearly vs daily bins, the results section seems a bit unstructured.

270: full vs. null, rather than “full null”?

Discussion

Although the general findings seem well considered, I find some of the claims in the Discussion questionable. First, you claim the repeatability is “high” but later state that the

data are comparable to an average repeatability value reported in a meta-analysis, which doesn't make it sound like your values are particularly high. Second, you claim the data represent a "sizeable proportion of the adult lifespan" but you have nowhere told us what this proportion is for the individuals in your data set. Third, you say that the "stable social phenotypes" you have documented are independent of life history stage, but since you only analyzed adults, this statement seems unwarranted.

I think the Discussion could be better organized, leading each paragraph or section with a statement that clarifies what the main point is. Instead, you tend to start with a result.

Appendix B

Max Planck Institute for
Evolutionary Anthropology

Patrick Tkaczynski
Department of Primatology
Deutscher Platz 6
Leipzig 04103
Germany
Tel: ++49 (0)341-3550-248
patrick_tkaczynski@eva.mpg.de
28th May 2020

Dear Dr Dunn,

Thank you for the opportunity to submit a revised version of our manuscript, "*Long-term repeatability in social behaviours suggests stable social phenotypes in wild chimpanzees*" for publication in *Royal Society Open Science*.

Following the comments of the reviewers we have made changes to the framing of our study and the interpretation of our results, which we feel have substantially improved the paper. In particular, instead of presenting the length and size of our dataset as meritorious in and of itself, more effort throughout the manuscript is made to highlight the value of chimpanzees to this field of research. Although the duration of observation of specific individuals is highly variable within our dataset, the addition of detailed control variables (rank, reproductive state etc.) allow us to accurately quantify just how stable social phenotypes are within one of our closest living relatives. In addition, we made a slight adjustment to the analysis as the first reviewer highlighted potentially issues with zero inflation in some of our models. These new analyses change some of the values, but not the effects previously observed and presented.

We have attached detailed responses to reviewers in bold below their comments. We look forward to feedback on the revision and thank you again for the opportunity to resubmit.

Yours sincerely,

Patrick Tkaczynski

Response to reviewers

This study compares repeatability measurements for three aspects of social behavior (rates of grooming (duration), aggression (count) and association/gregariousness (index expressing deviation from random expectation)) in male and female members of a wild chimpanzee population studied individually for periods up to several years. The stated aim is to use data from a long-term study to assess intraindividual variation in behavior over time, as well as interindividual differences that cannot be explained by time-varying situational or individual attributes, such as group size, age and rank. In the Introduction, the authors articulate several predictions based on their understanding of behavioral variation over the lifespan in the study species. Specifically, they expect (1) greater intraindividual variation in rates of aggression for males than females, (2) overall low repeatability for rates of grooming given (this prediction would benefit from some comparative – low with relative to what?), and (3) low repeatability for association (again, relative to what?).

Later (in Methods), it becomes clear that they also bin their data for the analysis in two ways, by day and by year, although the predictions articulated in the Introduction do not refer to these two bin sizes and one has to read the entire paper to understand the relevance of these alternative data-collation procedures. This comparison is interesting, however, and does have implications (not developed, but briefly mentioned in the Discussion) for interpreting analyses of individual differences comparatively. Differences in repeatability for the three behaviors, between sexes and between the two time-frames analyzed, are interpreted in a reasonable way.

Overall, this was an interesting paper based on rare longitudinal data in a highly social and very long-lived species. However, the presentation is quite complex (many comparisons, complex analysis), and the manuscript was a very difficult read. It would benefit from much tighter organization, with goals articulated more clearly in the Introduction, and retention of the organizational roll-out from the Introduction in the Methods (as much as possible) and in the Results and Discussion. Avoid using varying terminology. Be very careful with word choice. Specific comments below may help.

We thank the reviewer for the feedback and suggestions on structure and clarity. We provide specific responses on the restructuring in relation to comments regarding specific sections (Introduction, Methods etc.) below. Throughout the manuscript, we have implemented clearer and consistent terminology that should improve its readability.

Also, the data set itself should be better described in the text. Follows of chimpanzees under natural conditions are, I believe, always somewhat serendipitous, as these primates do not occur in stable associations. The authors do not describe how focal subjects were chosen (how randomly?), nor how the follows of individual subjects were distributed over time, which seems important if one of the motivations of the study is accurate assessment of behavior over time through the use of a long-term data set. Information on the data set is available in a supplementary table, but I would recommend summarizing key information in the text, relating especially to variation in the amount of data per individual, both in terms of regularity of observations and total time span over which observations occurred.

The reviewer is correct that chimpanzee follows can be serendipitous, however, Tai chimpanzees are quite distinctive among other populations in their levels of gregariousness. This factor, coupled with the fact that all groups are followed daily by a combination of field assistants, students and other researchers, means we are typically fortunate to be able to implement pseudo-randomisation of the order of focal follows and avoid individuals being sampled for consecutive days, or prior to other individuals being sampled within a working month.

We have made this clearer in the Methods section and have indeed moved our supplementary table into the main manuscript in order to be clear about the level of sampling of individuals over time.

Introduction

It would benefit communication if there was some reorganization of the introduction so that a reader knows earlier what this study will be about. The first several paragraphs are not particularly focused or well organized or written. I even think there could be more compelling "hooks:" you start out with social bonds and connectivity, but then at the end of the second paragraph we're onto cooperativeness (which might be related) but also gregariousness (which is different from bonding) and aggressiveness (which seems like something else altogether). You don't get to repeatability until the third paragraph. And not until after this, as you lay out predictions, does a reader realize that your main aim is to compare repeatability for different sexes and behavior types, and for data organized on a daily vs yearly basis. Make the conclusions one can draw from those comparisons the focus of the paper, right at the beginning.

We thank the reviewer for the structural suggestions and have reorganized the Introduction accordingly. Specifically:

- **We introduce repeatability immediately as a concept;**
- **We propose why repeatability in social behaviour, as evidenced in other species, is surprising and of interest to behavioural ecologists;**
- **Present mechanisms by which repeatability or apparent repeatability of social behaviour could manifest;**
- **How well these mechanisms have been explored in other studies;**
- **The advantages of exploring these mechanisms in chimpanzees;**
- **Our predictions.**

The introduction initially seems to describe the motivation of the study in terms of the necessity of covering enough of the organism's lifespan that one will avoid misinterpreting as persistent individual differences what might just be differences reflecting age, rank or demographic variables. This is a valid methodological point, but the text could do a better job making it convincingly. It would help to present (at least review) more specific information about the limitations of studies to date and highlighting any reports that compare a shorter vs. longer-duration data sets, and how conclusions differ.

I also think the methodological limitations of prior work is a secondary point (and therefore should be placed later than it is in the Introduction) to the main aim here, which is comparing sexes and behavior types (and time frames) in terms of the repeatability they exhibit. To achieve this aim, of course one wants to measure repeatability well, and that's where having a

long enough time span is crucial. This said, it is not entirely clear that your data are an improvement on other data: data from East group come from only a 4 year period, which is arguably not a very big chunk of a chimpanzee's lifespan. Again, more description of the data set will help to make your case.

We thank the reviewer for these two comments which have helped us re-frame our study. The studies we highlighted using short-lived species, or even studies over short timeframes in more controlled settings in captivity, do indeed provide valuable information about how stable social behaviours can be across different socioecological settings or life history stages. We now highlight this in lines (85-94).

Instead of presenting the length and size of our dataset as meritorious in and of itself, more effort here and throughout the manuscript is made to highlight the value of chimpanzees to this field of research. These animals use a diversity of social behaviours that are well linked to fitness outcomes. However, they also face diverse social challenges and settings over their long lives. Although the duration of observation of specific individuals is highly variable within our dataset, the addition of detailed control variables (rank, reproductive state etc.) allow us to accurately quantify just how stable social phenotypes are within one of our closest living relatives (lines 123-171).

Lines 58-59: State earlier what kind of social behavior differences you are referring to here. This sentence is really much more related to the subject of your report than the paragraph before, and a reader wants you to dig in here. The last sentence of the paragraph provides some details, but without references.

The Introduction has undergone substantial restructuring. This is now our opening sentence in which we highlight the range of social behaviours in which repeatability has already been investigated (lines xx-xx). In the Introduction we also make clear the behaviours that will be assessed in our study (lines 48-50).

Line 70-71 is a little confusing as you're trying to explain "consistent individual differences" (line 68) and yet you are invoking features of the individual that change over time (body size, health, dominance possibly).

Line 73 ff: it seems you are saying that one can, if data are limited, misinterpret APPARENT individual differences which are actually related changeable characteristics to age, rank, etc as being "true" or persistent individual differences. But if you write that the differences are "due to" these (changeable) features of the individual, then you appear to be contradicting yourself.

84: If the pitfalls of interpreting apparent individual differences from limited data are motivating your study, it would be useful to review this issue more: what kinds of time frames are common in the literature, have studies with longer durations reached different conclusions than those with shorter durations (ideally of the same organism)? How well have factors that might change over time been controlled in previous analyses? After reading the whole paper, however, it seemed that perhaps I misinterpreted what motivated your study – see comments at the beginning of this review. Even so, it would be helpful to say a bit more about adequate sample sizes for assessing repeatability.

We thank the reviewer for these three comments. This section now highlights the insights gleaned from former studies about the stability of social phenotypes across different settings in different taxa. We specifically want to highlight that there are multiple mechanisms that can lead to observations of stable social phenotypes, and accurate measures of their stability require taking into consideration the variation in socioecological settings or life history stages within a particular dataset (lines 72-84). We also highlight the advantages of different species for exploring these issues: certain species may have less diverse social behaviours, but can be observed across their whole lifespan or even have socioecological settings experimentally manipulated, whereas long-lived species, such as primates, may have a greater diversity of social behaviours that are easily observed but require substantial data collection effort over many years (lines 85-94).

94: does “social organization” really fluctuate? What exactly do you mean by social organization here?

This has been clarified as the “availability of social partners” (line 115).

99ff: Rank and “aggressive tendencies” seem to be viewed as the same thing here. I’m not entirely sure what “aggressive tendencies” are (later, 103, also referred to as “aggression”), but it need not be the case that the rate (per unit time) of aggressive interactions is correlated with rank in animal societies. If they are correlated in chimpanzee societies, that point should be made explicitly. And using precise language is very important.

105: shouldn’t you start a new paragraph when you describe grooming? And when you say “grooming behavior” please (again) be more specific: time spent grooming? number or diversity of partners? what exactly? Similarly, later you reference “association behavior” but what do you mean specifically? This too should be clear from the get go, to facilitate the reader’s comprehension.

We thank the reviewer for highlighting issues on clarity of language in relation to our behavioural measurements. We now clearly describe the variable being measured which has enabled us to also clarify our predictions for the variation in repeatability between grooming, aggression and association (lines 123-171).

Methods

The Methods section is very complex, and it is a challenge to present the information in a way that others can easily follow. The authors appear to use varying terminology which is not helpful.

124: Authors state that they include as subjects only those individuals who were sampled regularly, but more information is needed on the specific criteria: how regularly? How much variation in regularity? The description of data collection provides scant information and although the supplementary document would allow a reader to assess, it would be helpful to provide a little more basic summary in the text of the paper regarding how days and years are distributed over time for the individuals chosen as subjects.

Lines 187-195 specify the criteria for inclusion in either the daily or yearly analyses. For full transparency on the size and nature of the dataset, the supplementary table has now been brought into the main manuscript, detailing sampling effort for each subject in the study.

146: clarify what you mean by “cumulative” – is this simply a count of unique adult individuals with whom it associated in a given day?

Yes, this is what was meant by cumulative, but we have appropriated how you describe it here to improve our clarity! (line 244-255)

150: “summing their association” is unclear. What is “their association”?

We have provided more information on how this metric was calculated which we believe provides greater clarity for the reader (lines 247-269).

151: is this an appropriate standardization? If I am following, this would lead you to express the average association strength across all individuals in the subject’s community. But do we expect average values to be similar across communities when they differ in size? If chimpanzees prioritize being in association with a certain number of individuals at a time, for example, then the average association rate per possible partner must be higher in the smaller than in the larger community. Also, how does this standardization relates to the text at 153 ff?

We have restructured the Methods section to provide more clarity. The standardisation here captures whether an individual was more likely to be seen with many other group members given the group size and the distribution of party sizes. The index tells us that an individual was more gregarious than expected for an individual in that community in that year given the group size and party sizes.

We apologise for the confusion regarding the terminology in line 153, now line 259; although “standardised” is technically correct, it is confusing to mix this with the more commons statistical standardisation used in the rest of the analysis. Instead we write now: “association indices have to be standardized considered in the context of an appropriate null model”.

169: estimated with what degree of precision? How many subjects had ages estimated this way? How would uncertainty in ages affect results?

Of the 70 subjects, 39 had estimated ages, based on established methods for age estimation in wild chimpanzees (Reynolds, 2005; full reference in manuscript). We have included this information in our Methods (lines 294-295). As all individuals were definitely adults, we do not feel these age estimations would necessarily affect results, particularly as we include other variables such as reproductive state to account for changes in life history strategies.

177: so age was extracted for the beginning of the annual interval (171) and rank at the end that interval? This seems inconsistent. Why not compute an average age/rank throughout the year? Same for sex ratio (and is the sex ratio for adults only)?

We have moved the date for age and sex ratio calculations to the end of the annual interval for consistency. Calculating the rank at the end of the interval is more accurate than any of the other solutions for yearly values, as we use the Elo index which integrates interactions over the whole period.

193: but ARE interaction rates related to group size? This would not have to be the case. See comment about line 151 for a similar point.

We would contend that group size does influence interaction rates, particularly in the extremes of the group size distributions and in a species with a high degree of fission-fusion. In very large groups, low-ranking individuals in large parties would have limited competitive ability and either increase foraging time or reduce competition by occupying smaller parties. In both cases such an individual might have social time constrained. On the other hand, if you are in a very small group, you might not expect as many aggressions, because food competition is lower.

194-195: The text here is unclear as written, though I can see why for a repeatability analysis you'd want the standardization to be by group.

We have clarified that we used z-score transformation on group size within each group (lines 282-287).

196-198: This is also unclear. An offset would make sense if one were modelling counts (e.g. for aggression count), but how does it help if one is modeling rates (hourly grooming rate (line 216) and association index)? Would one expect the amount of observation time to influence the proportion of time spent grooming or the association index? Justify.

This was misrepresented in our description, as no offset term was included for these models. We changed the daily grooming models into negative binomial count models, and they do now contain an offset term for observation time, but the models using grooming rates or association indices do not have any.

199: group by year – do you mean “group-year”? Is this the same as “Year within group” referenced on line 210? Data collected from a given group within a given year may not be independent, but then data collected on Aug 30 are probably also not independent of those collected on Sept 2. I am not sure if there may be a better solution here to account for temporal clustering in the data in a series of successive time blocks.

We thank the reviewer for this consideration. As you rightly describe, there is not complete independence of data from one month to the next, but we are also constrained by the distribution of our data across time in how we can account for temporal autocorrelation. As such, although crude, given the data, our approach is likely the best

we can do. We have now ensured that we are consistent in how we report this random effect.

Also, here you say “radians of Julian date” and later you refer to sin and cosin... Please clarify how you account for seasonality and, more generally, be consistent in the terminology.

This is a proxy seasonality measure to account for any deviations from a uniform distribution over time the expression of a variable, here social behaviour. In the absence of detailed ecological data, this measure has proven a useful proxy in ecological studies (Stolwijk et al, 1999), including in several chimpanzee studies conducted within our research group (Wessling et al, 2018; Samuni et al, 2020; full references in manuscript).

208: why not take the group size on the same day? Why take a yearly average?

Group size was calculated on the daily level for daily models. We hope that this is now clearer in the manuscript.

208: why is group ID included as a fixed effect?

We only had three groups, which prevented us from including “group ID” as a random effect, which might have been preferable.

216: do you mean logit, not log (Warton and Hui 2011)? Was there zero inflation in the grooming data? Is a Gaussian model appropriate for % of time spent grooming? That is, do transformed values satisfy linear modeling assumptions? Proportions of continuous variables like time are often very tricky as they are bounded by 0 and 1. In general, have the authors conducted any model diagnostics?

We thank the reviewer for this comment. We had previously conducted diagnostics for multicollinearity but had not tested for heteroscedasticity and overdispersion in count models. There were no changes in the yearly models; we are using grooming minutes per hour, which is not bound by 0 and 1, because no individual can physically groom the whole day. For yearly values, we had a total of 2 cases of individuals who had 0 minutes of grooming in a year, so there was no problem with zero inflation. We used the logarithmic transformation to improve model fit, as the residuals diverged from normal distribution. Thus, nothing changed for the yearly models. For the daily models, zero inflation posed a problem for the male and female grooming models and the female aggression model. We changed models to negative binomial, with minutes of grooming as outcome and offset for observation time. Model diagnostics were improved by this, but overall results were not affected in a dramatic way.

Results

In general, I recommend reporting how many lines of data you had for each model, possibly per subject. That is, how many repeated time units occurred for N subjects?

Because there were no predictions made about analyses based on yearly vs daily bins, the results section seems a bit unstructured.

The results have been restructured. They now introduce the general results, that all three behaviours were repeatable at both levels of data aggregation, before discussing sex differences in repeatability for social behaviours. For clarity on sample sizes, we have now moved our Supplementary Table into the main manuscript. This lists for each individual how many days and years they had included in the daily and yearly level models respectively.

270: full vs. null, rather than “full null”?

Thank you for spotting this, this has been corrected.

Discussion

Although the general findings seem well considered, I find some of the claims in the Discussion questionable. First, you claim the repeatability is “high” but later state that the data are comparable to an average repeatability value reported in a meta-analysis, which doesn’t make it sound like your values are particularly high.

This is a fair comment; in the revised manuscript, we have used more careful language to show that we anticipated much lower repeatability estimates in chimpanzees compared to those observed in other species given the motivations for flexibility described in the Introduction (lines 158-171).

Second, you claim the data represent a “sizeable proportion of the adult lifespan” but you have nowhere told us what this proportion is for the individuals in your data set.

As previously described, we have reframed the manuscript to focus less on the length of the dataset and more on the within-individual variation in socioecological settings, intrinsic state of life history strategies represented in the dataset. We admit it will take many more years of data collection to effectively measure stability across the whole lifespan of long-lived chimpanzees. Nevertheless, our substantial dataset does allow us to measure stability over several years for our subjects, including several changes in socioecological settings (group sizes), and importantly, variation in intrinsic state or life history, such as changes in age, rank, reproductive state or strategy. We now include our Supplementary Table in the main manuscript to be transparent about the variation in observation time for each individual included in the study.

Third, you say that the “stable social phenotypes” you have documented are independent of life history stage, but since you only analyzed adults, this statement seems unwarranted.

In our revised Introduction, we present emerging evidence that life history and reproductive strategies vary within adulthood in chimpanzees. Now in the revised Discussion we make clear that the results pertain only to adulthood, but that variation in strategies can occur within this period in long-lived species such as chimpanzees (e.g. lines 447-448).

I think the Discussion could be better organized, leading each paragraph or section with a statement that clarifies what the main point is. Instead, you tend to start with a result.

We thank the reviewer for these suggestions on structure. The revised Discussion has been reorganized, we initial introduce the methodological insights from our study (lines 453-466), before exploring the differences amongst the three behaviours (lines 468-514), ending with how our results inform our understanding of the mechanisms underlying repeatability in social behaviour (lines 517-561).

Reviewer: 2

Comments to the Author(s)

The authors examine consistent between-individual differences in agonistic and affiliative social behavior in wild chimpanzees, using a dense longitudinal sample of individuals living in 3 distinct communities in the Tai forest of Cote D'Ivoire. The authors specifically look at repeatability in daily vs annual levels of aggression, party association, and grooming given. The authors find that individual chimps are generally consistent in their preferences to associate in parties of a particular size and the amount of time they spend in association, though on an annual level males were less consistent than females. Males were also less consistent than females in their tendency to give aggression and grooming.

Recent examinations of consistent inter-individual differences, such as this study, provide important insight into behavioral strategies that deviate from population averages, and that were historically considered noise surrounding species-typical behavior. They also lend important insight into the broader evolutionary and ecological significance of what are usually termed "personalities" in humans. What I believe would strengthen this paper is to contextualize the inter-individual differences better in terms of social strategies that are relevant to chimpanzee social life. I flesh out specific areas for improvement in the following line-by-line comments.

Line 36-38: The length of previous collection time periods alone does not necessarily warrant further research in consistent individual differences. There is quite a lot of emphasis on the size of data set in this study as a measure of its significance. While the data set is impressive for a long-lived animal and certainly hard-won, many species examined for repeatable behavior have been observed over similarly long periods relative to their lifespans (many insects, birds, and fish, e.g. 3 consecutive flocking seasons in great tits that live to 13 years at maximum, Aplin et al. 2015 Animal Behavior). Observation years per individual in this study ranged 6 - 12 years in this study, for an animal that lives perhaps up to 65. Observations in chimps are perhaps more consistent and even over each year – how might this make for particularly reliable and robust estimations of repeatability?

We thank the reviewer for these comments, and as stated in our replies to reviewer 1, fully admit that a focus on the length of studies in other species did not do justice to those studies, nor our own. Instead, we have revised much of the Introduction, and in particular this section. We now highlight the benefits of different species with different life histories in examining the repeatability of social behaviours.

As you say, these different species offer the opportunity to examine social behaviour across life history or varying socioecological settings. However, many of the studies only

focus on one form of social behaviour, and in piecing together the evolutionary history of stable social phenotypes, studies of wild great apes are currently lacking.

Our timespans of observations varies considerably between different chimpanzees, but this long-term dataset still allows us to explore stability in social behaviour across seasons, and for many individuals, across variation in individual characteristics such as reproductive state or dominance rank. The Introduction now gives a fair representation of studies on repeatability in social behaviour to date, and the insights that we can potentially offer using long-term data from wild chimpanzees.

Intro paragraph starting line 47: Again, what is the significance of repeatable behavior? Currently this paragraph highlights various internal and external stimuli that can cause behavior to vary. While this is true, an interesting aim is to determine whether behavior, which has elsewhere been determined to be adaptive (e.g. affiliation, coalitionary aggression), is flexible to the moment or representative of a more constant trait.

Line 91: I suggesting strengthening the justification for this analysis in chimpanzees. The authors seem to present 2 hypotheses in the introduction, either personality is canalized by early environmental density dependent conditions (social niche), or behavior varies by life history stage and/or socio-ecological condition. If these are two hypotheses the authors wish to test, how are fission fusion dynamics relevant?

We thank the reviewer for these two comments. In the final paragraph of the revised Introduction, we now try to relate the significance of stability in social behaviour in adult chimpanzees to these hypotheses you highlight. We provide more detail on why these behaviours should be quite flexible in a fission-fusion society, such as that of chimpanzees, on a day-to-day basis, and explanation of why flexibility over the adult lifespan might be expected based on variation in life history and reproductive strategies. Therefore, if stability rather than flexibility is identified in the chimpanzees, it lends strong support to the hypothesis that these social phenotypes are canalized during ontogeny or early life.

Line 145: Clarify, does association mean spatial association/party membership excluding time grooming, or do these 2 measures overlap?

In lines 243-269 we have expanded our explanation of the association variable on the daily and yearly level. This daily measure is a count of the unique adult individuals with whom the focal associated on a given day. The yearly measure is a social network measure, i.e. the strength of an individual within a social network based on dyadic association rates (how often the dyad were in the same party).

Line 153: I am confused as to the kinds of null model permutations the authors conducted. Here the authors state that association indices must be measured against a null model. However given the dyadic non-independence of social interaction, all social measures should be modeled against random expectations. It appears later that they may have correctly compared models of all behavior types to null models - line 225. Please clarify.

Our yearly association measure is a social network metric. An individual's position within a social network is determined by the emergent properties of the network, therefore, all individual's metrics (here association strength) are not independent of each other, they are in fact, defined by each other. Therefore, to address this, we ran permutations to randomise the networks. This allows us to compare whether association strength for individuals was completely random, or in fact, association strength was non-random due to within-individual association preferences. We have provided further clarification on this in lines 257-260.

Paragraph start line 164: I appreciate starting the statistical analysis section with an outline of the number of models to keep track of.

Thank you.

Lines 176 – 177: Clarify here over what time frame annual rank was calculated. Currently it sounds like it was measured on a single day, August 31.

We have clarified when rank was extracted for the yearly and daily binning of data in lines 300-305.

Lines 188 – 195: I am unclear regarding why measures such as sex ratio, number of partners available, and group size are to be included in the repeatability model, when association permutation arguable already controlled for them.

Please see our comment above regarding the permutations. Controls on group-level dynamics such as sex ratio still need to be included in these models.

Line 221 – I suggest using consistent terminology and choosing either “association” or “gregariousness” to use throughout. Both can be used when introducing the meaning of association. Also line 260.

Thank you for highlighting this; the terminology is now consistent throughout.

Line 233 – Suggest also citing Nakagawa and Schielzeth et al 2010 Biological Reviews.

Thank you, this has been added.

Table 1 – Good, clear layout of results.

Many thanks.

Fig 2 & 3 – Please clarify: the x axis represents how consistent an individual was in its behavior over time, and red indicates how much that individual interacted relative to community average. Is it a coincidence then that all individuals' that interact less than the population mean are also low on repeatability?

We added axis labels to the plots and additional information to facilitate interpretation. The mean for each individual is the deviance of the random effect from the overall mean, plotted with the prediction interval based on within-individual variation. There was no consistent correlation between the variance and the mean; individuals with large variation would, if anything, cluster in the centre of the distribution due to regression to the mean.

General comment: Does individual level repeatability correlate w years/days observed? This would not be damning if so, but could highlight a limitation of the approach.

In our analysis we have not specifically tested individual repeatability; the repeatability coefficients represent the amount of variation in the population attributed to inter-individual differences. However, we do discuss how data aggregation can affect results, with implications for effectively sampling of individuals to accurately measure repeatability.

Line 310: Add comma after “and”.

Thank you, this has been added.

Line 313: What are the many former studies? Cite.

We have rephrased this sentence to focus on how controlling for multiple factors is important for interpreting the results of our study, rather than something that sets our study apart from others (line 443-451).

Line 318 - 326: This current framing of significance is weak. Humans are the primary subjects of all personality research. It is not surprising that their closest evolutionary relatives also show repeatable behavior. What do differences in repeatability on annual vs daily scales and between males and females mean about chimpanzee social strategies? I suggest the authors get straight to this meaty interpretation, particularly in paragraphs starting lines 327, 332, 345 & 356. Each of these paragraphs currently leads like a summary of statistical results. I suggest leading with a description of differential power structures and seasonality in males and females, and then tying them to the results in terms of how they would shape within-individual variation in aggressive/friendly behavior.

We agree with the reviewer and have restructured the Discussion accordingly. We initial introduce the methodological insights from our study (lines 453-466), before exploring the differences amongst the three behaviours (lines 468-514), ending with how our results inform our understanding of the mechanisms underlying repeatability in social behaviour (lines 517-561).

Line 324: Thompson Cords 2018 Ecology and Evolution also calculate repeatability in grooming in adult female blue monkeys.

We have added this reference, many thanks.

Line 355: This idea about constrained preferences and its contrast with flexible choice sounds interesting. Please develop it further and introduce it earlier on.

We thank the reviewer for highlighting this and have taken the opportunity to expand on this when discussing methodological considerations in data aggregation early in the revised Discussion (lines 453-466), and then expanded on this in our discussion of the association and grooming results.

Paragraph starting line 367: It's unclear what this paragraph is trying to achieve. Is it setting up a future study? Or is it tying the findings into the original discussion of the social niche hypothesis in the introduction? Some kind of return to and evaluation of that original hypothesis would be valuable. Your introduction seemed to lay a promising groundwork for a comparison between social tendencies arising from social niche specialization and/or being temporary life stage strategies. Could you speak to one or both ideas more and what your results mean in relation to them?

We thank the reviewer for the suggestion and now use this final section to describe how our study can inform on proposed mechanisms underlying repeatability in social behaviour.

Line 338: Replace "generated" with "characterized".

This has been done.

Reviewer: 3

Comments to the Author(s)

This study investigates the long term repeatability in social behaviors in wild chimpanzees. This repeatability could suggest stable social phenotypes. The study resulted in a dataset of many individuals and data covering more than 20 years (however, the individual mean is only 6 years and the maximum 15 years). During a short period, social bonds adapt because of the instability of dominance hierarchies, fluctuation resources availability, individual states etc., adaptation needs flexibility. Besides, there is a consistent individual difference in social behavior in a various range of taxa in a long period of time. There is a stable tendency in interactions. The degree of repeatability is linked to genetic and stable adaptations due to the experiences of the individual during his development. The social strategies of each individual are based on their own characteristics. Long term study are interesting to show individual difference independent from a special step in life cycle. It permits to see if the social behavior is reproducible over a long period of time.

I read this manuscript favorably and believe this is worth publication, but have also some concerns.

We thank the reviewer for the positive feedback and believe the revised manuscript should address the concerns raised.

My biggest concern is the time frame. The terms of analyzed data from the 45 individuals varied from 3 years to 15 years (mean = 6 years). I could not understand why this is so short considering their long life span (>50 years) and the studies' length (20 years). 6 years are not enough to see the chimpanzees' life-long stability. The description of the abstract "Using data spanning over 20 years, we demonstrate that multiple social behaviours are repeatable over the long-term in wild chimpanzees" is misleading.

We agree with the reviewer that our original focus on the size and timeframe of the dataset was not appropriate. We have now much revised the manuscript to highlight the insights that can be gained from different species and different datasets to address how stable social behaviours are across multiple contexts. As part of this, we now make a more convincing argument for studying the stability of social behaviour in wild chimpanzees that does not depend just on the duration of the dataset. We now do not argue that we assess a substantial proportion of the lifespan of our subjects, but that with such a dataset we can observe whether their social behaviour changes over meaningful transitions, such as changes in rank or reproductive state.

Grooming is an important component of social bond formation. The contingent grooming choice is based on a wide range of parameters such as audience, partner rank or context, as reconciliation after aggression for example. Grooming is used to reach social goals as dominance rank and formation of social bonds which have a huge influence on fitness. For the grooming they extracted the time focal individuals spent grooming adult partners. They specified "we focused on grooming given to others rather than overall time grooming, i.e. including grooming received, as this would reflect a tendency to attract grooming partners rather than an individual tendency to groom". This explanation is not very clear to me. Does it mean that they did not use mutual grooming? Yet mutual grooming seems important to measure the strength of social bonds. This point may need further clarifications.

Our measure includes all grooming the subject gave to conspecifics, including during mutual grooming, and within polyadic grooming clusters. We have clarified this in lines (238-240).

Social trends seem to be important. Understanding how some individuals become more aggressive, affiliated or gregarious, than others, requires further empirical explorations. However, in the discussion they only mention the immature period during which young chimpanzees are systematically linked to their mother. Much of the discussion is focused on this specific point. It might have been interesting to raise other factors. The social niche hypothesis suggests that coherent individual behavioral differences occur due to the specialization of the niche to improve intra-species and/or intra-group competition for resources. However this theory is mentioned only briefly without any explication. I think this could have been deepened.

We thank the reviewer for the suggestion. We have restructured the paper to introduce the social niche hypothesis and other mechanisms that could lead to consistent individual differences more clearly in the Introduction (lines 72-84). In the Discussion, we now describe how our results lend support to either heritable factors contributing to variation in social phenotypes, or a canalization of social phenotypes during development, such as through the social niche specialisation (lines 517-546). We also

now highlight here and throughout the manuscript that life history strategies do not remain constant throughout adulthood, and that further analysis of adjustments to social trends would be valuable to this field of research (lines 547-556).

Appendix C

Reviewer comments to Author:

Reviewer: 1

Comments to the Author(s)

This revised manuscript is a marked improvement over the previous version. The introduction gives a clearer lead-in to the study, and it is much easier for a reader to follow the methods and arguments. However, I still think the authors can improve the organization, especially of the Discussion. It would help a lot if they followed the roll out of predictions at the end of the Introduction in organizing the Discussion, i.e. use the same predictions in the same order in both sections of the paper. It seems there are three natural sections: comparisons of Repeatability to other reports/taxa/behaviors, comparison among the three types of behavior examined here, and comparisons between the sexes (for each of the three behaviors). Comparisons between the two data aggregation scales might be a fourth. Set up a parallel structure when discussing these comparisons so that the Introduction and Discussion mirror each other in terms of organization/structure (at least, for discussing these particular results). Make sure the Discussion states explicitly whether expectations were met or not.

I also found the Discussion a little longer than I think it needs to be, especially the very last section. Perhaps this can be reduced a little. These are interesting questions but the data really cannot address them.

I am still concerned about two aspects of how the dataset was put together. For the yearly data set, why take one single daily value (Aug 31) to represent an entire year (instead of averaging across days of the year? I don't see how it can possibly be true that the value on one date is a better representation of the "whole year" than some kind of average (median, mean). The authors have also not justified why it is ok to accept follows that are only 3 hours long as representative of a "dawn to dusk" follow: does a follow of this length represent an accurate assessment of the behavioral variables used in the analysis, especially when data are collated on a daily basis? The authors need to make their case here, or possibly swap out some measurements. The latter would be a bigger deal, I realize, as reanalysis would be required.

We thank the reviewer for the positive feedback on our revision and for comments on how to further improve the manuscript.

Regarding your points on the analyses, we have now re-run the models using average rather than year-end values for factors such as rank or sex ratio. While this does not change the results significantly, we agree that calculating the variables this way is more appropriate.

On the Methods, we also now clarify the data collection more clearly, providing more detail on the number of hours per focal observation (means and standard deviations).

For the Discussion, we have reorganized its structure in line with the reviewer's suggestions and have shortened the final section. Throughout the Discussion we now highlight whether our predictions were met or not when discussing the implications of the results.

We have responded in more detail regarding each of these points when raised in the line-by-line comments below.

Line by line comments: most relate to expressing things correctly or more effectively. A few are

other sorts of questions.

58: I would say “often” rather than “generally”. So few species have been examined, it’s hard to know if it’s really general.

This has been changed (line 58).

60: “evidenced repeatability” is not correct English (to evidence is not a verb)

This has been changed to : “Where observed, repeatability in social phenotypes...” (line 60).

63: “vary” would be better than “fluctuate” (see also 115 for noun form, and several other cases in the manuscript – I think VARY sounds better in ALL of them)

We have checked for occurrences of “fluctuate” or “fluctuations” and changed throughout to “vary” or “variation(s)” (e.g. lines 62, 65, 125).

78: typically one does not use “etc” in formal writing

This has been removed.

95: Does everything have to be a “model” species? Personally I don’t think any primate is a model species: that term is usually refers to the lab mouse/rat and their ilk. Chimpanzees and primates are seldom model species because they are way too hard to work with. How about “useful” or “appropriate” or “interesting”? I think it’s a plus that this paper is NOT about a model species!

We have changed “model” to “interesting” in light of the reviewer comments (line 95).

99: do you mean “life history STAGE” here?

Yes; this has been corrected (line 99).

107ff: it is convention to use past tense in writing about one’s own study... many changes needed

We have corrected any use of present tense used within the manuscript (e.g. lines 107, 110, 127).

118: “affect”, rather than “impact on”

This has been changed (line 117).

120: word missing? to BE flexible?

This has been corrected (line 119).

122: consider giving SOME idea of what “longer term” means here – years?

We have added “i.e. from day-to-day or year-to-year” (line 121).

125: replace “the composition of bystanders” (unclear) with “which bystanders are nearby”; where you refer to “rank differences with available partners”, presumably you mean differences in the subject’s rank relative to different partners, so just refer to partner rank, not rank differences?

We have made the suggested changes (line 124-125).

132: “our other behaviors of interest” – a reader doesn’t know what you mean here... perhaps add (see below)? Also, no reason to use the possessive.

We clarified the behaviours and added “(see below)” (line 130-131).

136: “their” technically refers to “ranks” (the last plural noun) ... rewording needed here

We have changed this to “Although females do change rank,, female hierarchies are comparatively stable” (lines 135-136).

144-146: this sentence needs some rewriting – better not to use future tense, and in general it’s confusing, possibly some words missing

This has been changed to: “Furthermore, male, but not female, aggression rates vary with mating competition (28,63), leading to lower repeatability of aggression in males compared to females.” (line 143-145).

149: adjust –explain with a brief reference that the adjustment is carried out by choosing which sized party to join or remain with

This has been changed to: “As has been highlighted, chimpanzees sociality is characterised by a high degree of fission-fusion; this allows individuals to adjust with whom they associate (either specific association partners or specific party sizes) dependent on variation in within-group competition arising from ecological constraints, such as the availability of receptive mating partners or food (47).” (lines 147-151).

155, 501: extant?? are you trying to say the predation pressure is high?

Tai chimpanzees are one of the few chimpanzee populations that have existing natural predators within their range, and this existing predation pressure is suggested as a reason for these chimpanzees being highly gregarious. However, for clarity for the broader readership we have simply removed “extant”.

159: which “studies to date”? all of them? More importantly, it’s not clear why you expect this. Did other studies include more variable life stages (not only adults)? You go on to describe how many things change both for males and females during adulthood, so this text seems to argue AGAINST the idea that limiting the study to adults should lead to an expectation of low repeatability. I was left a bit baffled.

We agree with the reviewer that this opening statement for the paragraph is confusing. We have removed it and in doing so, the paragraph is now focused on the implications of identifying repeatability in our population and for our chosen behaviours.

175: here you reference “nest to nest” focal follows, but then later you say the follows had to last (only) 3 hrs... this seems inconsistent. Did all the short (not full day) follows start at a night nest in the morning? Are there diurnal rhythms of activity, especially the behaviors you examined, in chimps

and if samples were biased by time of day (all started early, fewer afternoons represented), isn't that a concern? Explain for the reader.

Even though the goal of assistants and researchers is to follow the chimpanzees from nest to nest, and this is mostly achieved, it is not always possible: typically this is because the focal is lost during the course of the day. In that case, observers will choose a different individual to follow, but not be able to achieve a 12h follow period. We have chosen 3 hours as a cut-off value because it should represent enough time for individuals to show the types of behaviour of interest here. The vast majority (81%) of focal samples were in excess of 8 hours. This information has been included and we have added the following information in the manuscript: 'The three hour cut-off value was chosen as researchers would sometimes lose chimpanzees during follows and changed the focal individual. Changes of subjects did occur at different times of the day, so shorter observations would not be biased towards specific behaviour.' (lines 188-196)

Table 1: For the daily analyses, there appears to be no information about how the individual days were spread over time. Can you please provide this information or clarify?

We have added a column for each individual indicating the first and last years in which they were included in the analysis.

238: "other individuals" is still only adults, right?

Yes, that is correct. Information added.

241: what kinds of behavior were included as aggressive?

We included both non-contact and contact directed aggression. Information added (line 242-245)

254: omit apostrophe

This has been deleted.

261: by CHANCE, not by random

This has been changed (line 264)

263: what do you mean by "subsequent parties that originally had the same party membership" – what is a subsequent party? Not quite following here.

This has been changed to "We conducted permutation analyses to confirm that associations were different than would be expected by chance. For this analysis, we generated 1000 permutations of observed parties in which the number of individuals per party was kept constant; subsequently generated parties that had the same party membership as the observed party were included in the analysis to account for autocorrelation (78)" (lines 264-267).

278: why would sex ratio not be calculated as the average across all days of the year?

As all other variables, sex ratio is now the average values for the year in question.

283: in, not into. For yearly group size, when do you measure this? Presumably group size may change over the course of a year, so is it a time-averaged group size across the days of the year?

We have now changed the data to reflect the mean group size across the year.

284: based on, not based

This change has been made.

292: even those who were not yet adult at the beginning of habituation had estimated ages, right? If you don't KNOW their date of birth, you must be estimating their age: you just have more to go on in these cases, as you witness them growing/changing more than if they were already adult at the start of habituation.

This has been corrected to: "The age of all individuals was either known (for individuals born during or post habituation) or was estimated in the beginning of the habituation period by experienced observers using established indicators of aging in wild chimpanzees (83)" (line 296-298)

296: why do you assign age at the beginning of the year instead of the year's midpoint? Why is age assessed at the beginning of the year whereas sex ratio (and rank?) is assessed at the end of the year?

As age is z-standardized, there is no difference between the age variable taken at the beginning or end of the study period. However, for consistency, the age is now established at the end of the year.

300: A reader should not have to read additional papers to understand what you did here. Please provide a little more information: what kind of modification was made, were all pant grunt interactions between adults only, both sexes together? Finally, if rank changes, especially for males, I think using a rank on ONE day of the year needs strong justification: why not take the average Elo rating for the individual across the year?

We have provided more detail on the Elo methods (line 303-310). We have now changed the rank variable for the yearly analyses to reflect the mean rank for that individual for that year.

314-315: are these separate variables or different levels of a single categorical variable? During a year, a female might have a newborn AND unweaned offspring (the newborn grows)... which takes priority then?

We thank the reviewer for highlighting this. We have clarified that is a categorical variable consisting of three levels. As with the daily measure, the presence of a new-born was prioritized over the presence of un-weaned offspring in this categorization. This has been added in the method description (line 314-326).

336: you have not yet described cosine and sine functions?

These are mentioned on line 291-292, but we have now clarified here that the "radians of Julian date" refers to the sine/cosine functions.

341: so does this rate vary, in principle, from 0 to 1? Clarify.

The grooming rate is from 0 to however many minutes per hour of grooming occurred; this has been clarified (line 351-356).

348: unique individuals? or average # individuals in the party/parties?

Unique – we have clarified this (line 357).

365: you mean EXAMINED, not ESTABLISHED, I think

We have changed the wording.

382: instead of “there were few differences” say “differences were minimal”

We thank the reviewer for the suggested wording and have implemented it.

Table 2: The legend should explain the column headers more (perhaps move some info from the text to the table, or repeat it briefly)

We have revised the table to include definitions for the different variance measures and to differentiate table columns referring to full null comparisons or correlations between the different data aggregations.

Fig. 1: maybe jitter the symbols for grooming, or indicate in legend that male and female values coincide and are superpositioned. Figure legend: “delineated” is not the right verb. I think this legend could be written more clearly – explain what R^2 is, briefly.

We thank the reviewer for the suggestions. The plots are no longer superpositioned upon each other following using average rather than year-end values in the analysis. The legend now reads: “Figure 1: Repeatability of Chimpanzee Social Behaviours. Overview of the effect sizes attributed to the individual random effect, depicted by behaviour, sex, and timeframe (yearly vs daily) over 20 years of data. R^2 refers to individual R^2 , i.e. the difference between conditional R^2 (combined variance of fixed and random effects) and marginal R^2 (variance of fixed effect only).”

Fig 2-3: legend includes phrase “given all fixed and random effects” – this sounds a bit odd. Also avoid having the word “given” twice in a sentence.

This has been reworded in each legend to: “Blue individuals: higher than average expression of the variable of interest (grooming, aggression, and association), accounting for all fixed and random effects within the model, red individuals: less than average expression of the variable of interest.”

439: I believe one should not use behaviors as a plural noun.. maybe “types of behavior” or “behavior types”? This is an issue in multiple places later as well.

We have used the reviewer’s suggested “types of behaviour” throughout the manuscript.

458: allows ONE (add the word “one”)

This has been added.

472: differences EXTEND (not extends)

This has been corrected.

472-473: why do you say they should influence fitness more? Don't require the reader to examine 3 additional papers. Are you speaking only about chimpanzees or is this intended as a broader statement?

This sentence and section has been reworded: "Our study shows that consistent individual differences in social behaviour extend to patterns of aggression and grooming. Both aggression and grooming involve direct, typically physical interactions with other group members. In chimpanzees and other species with similar social behaviour and structures, aggression and grooming influence rank acquisition (56,85,88,104,105), disease transmission (23,24,106), social bond formation and maintenance (49,51,54,65), and group cohesion for territorial defence (52,64,107,108). (lines 467-473).

479: this statement directly contradicts 507

Thank you for highlighting this discrepancy. We now clarify the importance of group cohesion with the following sentence: "These factors can have direct and indirect fitness implications in chimpanzees, e.g. frequent incursions from neighbouring groups is associated with reduced infant survival and thus reduced adult reproductive success (109)." (lines 471-473).

488: what is meant by "physically compete"? You mean directly, i.e. aggressively?

Yes, we have clarified that this means aggression and physical displacement of other individuals.

489-490: this sentence needs work, seems like words are missing or the lack of parallel construction just makes it hard to follow

This has been restructured into two sentences: "The contrast between daily and yearly levels of repeatability was strongest in female aggression, with much higher repeatability estimates in the yearly versus daily data aggregation." (lines 516-517).

523: life history STRATEGIES? What do you mean by this?

This has been removed.

543: this is a dangling modifier: the chimpanzees didn't reveal

This sentence has now been removed.

549: years ARE required, not IS required

This has been changed.

551: data ALLOW, not allows

This has been changed.

554: This what?

To make the Discussion more concise, this section has been removed.

Reviewer: 2

Comments to the Author(s)

The authors have improved their introduction greatly, with a much better focus on the potential significance of repeatable behavior in chimpanzees. The study is set up in the last paragraph of the introduction with clear hypotheses and predictions. In Methods, the reason for permutation methods for significance testing is clear now, as association strength is a social network measure. In Discussion, the authors take the appropriate room to discuss different time frames for aggregation.

We thank the reviewer for their positive comments and feedback throughout the review process.

The following are my remaining concerns by line number:

Line 154 – Guide the reader as to what “more gregarious” means – each individual at Tai spends a larger amount of time, on average, in a social party than chimpanzees at other sites do?

We have added in parenthesis the following clarification: “Furthermore, Tai chimpanzees are considered one of the more gregarious chimpanzee populations (i.e. individuals are likely to associate with all other group members relatively frequently, even within a day), likely as a consequence of low population density, high food resource availability and/or predation pressure (67–70).” (line 154-156).

Lines 243 – 269. Still unclear whether spatial association excludes time spent grooming.

The association measure does not exclude time spent grooming, and the two measures are a priori independent from each other. On the daily level, the association measure is the number of unique individuals with which the subject was observed associating, and therefore, is independent of time budget allocation (i.e. what they did while they were associating with others). On the yearly level, our association measure is a network measure, with each subject’s value an emergent property of the overall network, again making it independent of time budget allocation measures such as our grooming values.

Line 440-442 – The opening of your discussion would be stronger by stating the significance or biological meaning of your results coming near the meta-analytical $R = 0.32$ (note, I believe this should be “R” and not “R²”). Your last sentence of this paragraph seems to be the big takeaway and I suggest moving it towards the beginning of the paragraph.

We thank the reviewer for the suggestion and have moved the first sentence to the beginning of the paragraph, improving the flow of this section. We also changed R^2 to R.

Line 468 – I recommend starting this paragraph by stating the new idea to discuss, rather than reiterating your result.

The paragraph now opens: “Our results highlight the impact of different temporal levels of data aggregation in repeatability analyses, with implications for future research in this field.” (line 506-507).

Line 517 – I suggest just calling their social groups “fission-fusion” instead of “complex”.

This sentence has been removed in the revision.

Line 517 I suggest that at the beginning of the section “Causes and Consequences of Repeatable Social Behaviour” you bring the reader back to the last paragraph of your intro, where you had

competing hypotheses. Highlight that you've found evidence more in favor of one than the other, e.g. "Given repeatability in social behavior, independent of factors related to environmental and physiological conditions, we find preliminary support for early life canalization of social phenotypes..." Do acknowledge that there may be other variables that you didn't control for that could constrain social behavior during adulthood. This acknowledgment of limitations would more reasonably present early life canalization as not the definitive cause of repeatability but one worthy of further exploration.

We again thank the reviewer for the suggestions regarding the last section of the manuscript. We have restructured the Discussion so that it is more aligned with the predictions in the Introduction, reporting that we find support for either canalization of phenotypes during development or the possibility that heritable factors might explain adult social phenotypes. In general, we have made the Discussion more succinct based on your suggestions.

Paragraphs starting line 531 and 547

While I appreciate that the authors are probably setting up their future study on social niche specialization, the 2 paragraphs dedicated to this as **the** origin of repeatability (lines 531-556) can be abbreviated possibly to one paragraph. Currently, it seems like too much space and thought is dedicated to a topic that is actually somewhat peripheral to the paper.

As with our previous comment, we have reduced this section of the Discussion.